# PolarDepth: Monocular Transparent Object Depth from Polar-Physics Priors

**Wen Dong** [1 2]   **Haiyang Mei** [1 3]   **Yinglian Ji** [1]   **Zijun Zhang** [1]   **Wenyuan Zhang** [1]   **Pengwei Luo** [1]   **Bo Dong** [4]
**Shengfeng He** [2]   **Xin Yang** [1]

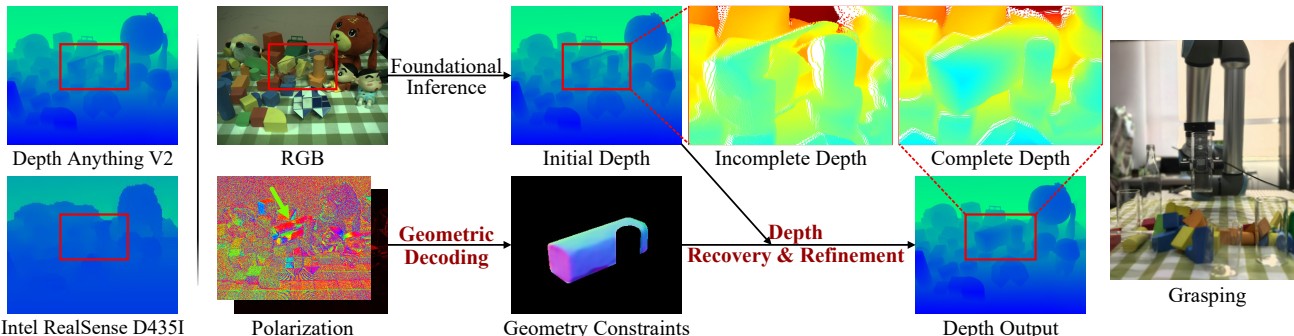

*Figure 1.* Transparent objects challenge conventional depth estimation, leading to failure in both learning-based methods like Depth Anything V2 (Yang et al., 2024) and active sensors such as Intel RealSense D435i. In contrast, polarization cues provide reliable geometric patterns even in transparent regions. The proposed PolarDepth extracts physically grounded geometric priors from these cues to recover accurate and complete depth, enabling precise robotic grasping and robust transparent object understanding.

## Abstract

Depth estimation for transparent objects remains a fundamental challenge, as RGB-based cues often fail in regions affected by refraction and light transmission. Polarization provides physically grounded information related to surface orientation and material properties, offering reliable geometric cues even in the absence of texture. In this work, we introduce PolarDepth, a monocular framework that incorporates both RGB and polarization inputs, including the degree and angle of linear polarization (DoLP and AoLP), to estimate dense depth and localize transparent regions. PolarDepth injects polarization-derived physical priors by estimating the refractive index, zenith angle, and azimuth angle from polarization measurements and embedding them into an implicit geometric representation that constrains depth inference in ambiguous transparent regions. To support model development and evaluation, we introduce PTOD, a dataset with synchronized RGB, polarization, and depth data and manually annotated transparent region masks. Experimental results demonstrate that PolarDepth achieves state-of-the-art performance in transparent object depth estimation. The findings highlight the effectiveness of embedding polarization-derived physical priors into learned representations for robust perception in complex visual environments. Code and model will be available at: https://github.com/wind1117/PolarDepth.

## 1. Introduction

Accurate depth estimation for transparent objects remains one of the most persistent challenges in computer vision (Liang et al., 2023; Ramirez et al., 2025; 2022). Most conventional methods rely heavily on appearance cues such as texture, shading, and specular reflections (Yang et al., 2024; Liu et al., 2025; Wen et al., 2025). While effective for opaque surfaces, these cues become unreliable in transparent regions, where light is refracted or transmitted rather than diffusely reflected. This breakdown violates the assumptions underlying learning-based depth predictors and often results in significant ambiguity (see Figure 1, left). Depth

---

[1]Key Laboratory of Social Computing and Cognitive Intelligence (Ministry of Education), Dalian University of Technology, Dalian, China [2]School of Computing and Information System, Singapore Management University, Singapore [3]Show Lab, National University of Singapore, Singapore [4]Cephia AI, INC. America. Correspondence to: Xin Yang <xinyang@dlut.edu.cn>.

*Proceedings of the 43rd International Conference on Machine Learning*, Seoul, South Korea. PMLR 306, 2026. Copyright 2026 by the author(s).

estimates may incorrectly assign background depth to foreground pixels or entirely miss object surfaces (Weibel et al., 2023; Bartolomei et al., 2025). Such limitations undermine transparent object perception in safety-critical applications, including robotic manipulation, augmented reality, and autonomous systems (Sajjan et al., 2020; Chen et al., 2022; Mei et al., 2025).

Polarization imaging provides an alternative source of information that is physically grounded and complementary to RGB appearance. Unlike intensity values that vary with illumination and texture, polarization encodes intrinsic properties of light reflection at dielectric interfaces, directly relating to surface orientation and refractive index (Fukao et al., 2021; Shao et al., 2023). Specifically, the degree of linear polarization (DoLP) is influenced by the surface zenith angle and refractive index, while the angle of linear polarization (AoLP) correlates with the azimuth angle (Miyazaki et al., 2004; Wu et al., 2025). These polarization cues remain informative even in regions lacking texture or color contrast, offering reliable geometric signals in scenes where RGB-based methods fail (Kalra et al., 2020; Mei et al., 2021).

Recent attempts to combine polarization and RGB for depth estimation have shown promise (Tian et al., 2023; Ikemura et al., 2024), yet many of these methods treat polarization as a generic visual inputs similar to RGB, extracting appearance-level features without explicitly modeling the underlying physics. As a result, the deterministic geometric constraints encoded in polarization measurements are largely underutilized.

To address this gap, we propose PolarDepth, a monocular transparent object depth estimation framework guided by polarization-derived physical priors. PolarDepth exploits the intrinsic relationships between polarization and surface geometry by decoding refractive index, zenith angle, and azimuth angle from DoLP and AoLP measurements. These quantities are embedded into a geometric representation via *Polarization-Guided Geometric Decoding* and *Depth Recovery & Refinement* modules. By grounding depth inference in physically meaningful constraints, PolarDepth resolves ambiguities in transparent regions and enables accurate, complete depth reconstruction even where RGB information is limited or misleading (Figure 1).

To support training and evaluation, we construct PTOD (Polarized Transparent Object Dataset), a novel RGB–polarization dataset comprising 2,878 scenes with pixel-aligned RGB images, AoLP and DoLP maps, ground-truth depth, and manually annotated transparent region masks. Experiments demonstrate that PolarDepth achieves state-of-the-art performance in transparent object depth estimation and segmentation while maintaining real-time inference with a compact ViT-S backbone, underscoring the effectiveness of polarization-guided physical reasoning.

Our main contributions are threefold:
- We propose PolarDepth, a monocular depth estimation framework that leverages polarization-guided geometric priors to address depth estimation for transparent objects, where conventional RGB-based methods often fail.
- We introduce a physically grounded decoding strategy that explicitly decodes surface geometry from polarization measurement and selectively injects physically meaningful constraints into ambiguous transparent regions.
- We present PTOD, a new dataset with pixel-aligned RGB, AoLP, DoLP, depth, and transparent region annotations, designed to advance research in polarimetric transparent object perception.

## 2. Related Work

**RGB-based transparent object depth estimation.** Existing methods can be broadly categorized into depth completion and direct depth prediction. Completion-based methods such as TODE-Trans (Chen et al., 2023) and DREDS (Dai et al., 2022) use transformers or simulated sensor data to complete missing depth. Others like LIDF-Refine (Zhu et al., 2021) and TransparentNet (Xu et al., 2022) reconstruct geometry from RGB-based depth estimates. However, these techniques assume that transparent objects exhibit reliable visual features, which is often not the case. Optimization-based methods such as ClearGrasp (Sajjan et al., 2020) leverage semantic segmentation and normal estimation to refine depth maps, but their performance depends heavily on upstream predictions. MODEST (Liu et al., 2025) jointly estimates segmentation and depth, yet it still struggles when RGB cues are weak or missing. GW-Depth (Liang et al., 2023) targets flat glass walls and additionally relies on glass boundary priors during training. NeRF-based frameworks (Dai et al., 2023; Duisterhof et al., 2024) represent transparent objects using implicit volumetric fields with multi-view input. Although recent work has improved runtime and generalization, these methods often require synthetic data and controlled lighting. Overall, reliance on RGB input remains a fundamental limitation across existing approaches.

**Polarization-guided vision tasks.** Polarization captures the directionality of light wave vibrations, which correlates with surface orientation and material properties. Unlike RGB data, polarization remains robust under degraded visual conditions. Prior work has applied polarization to semantic segmentation (Mei et al., 2022), material classification (Dong et al., 2024), and reflection separation (Lei et al., 2020; Li et al., 2020). It has also supported shape-from-polarization (SfP) (Ba et al., 2020; Lei et al., 2022). For depth-related tasks, methods such as DPS-Net (Tian et al., 2023) and PPFT (Ikemura et al., 2024) incorporate polarization into depth estimation, but often treat polarization images simi-

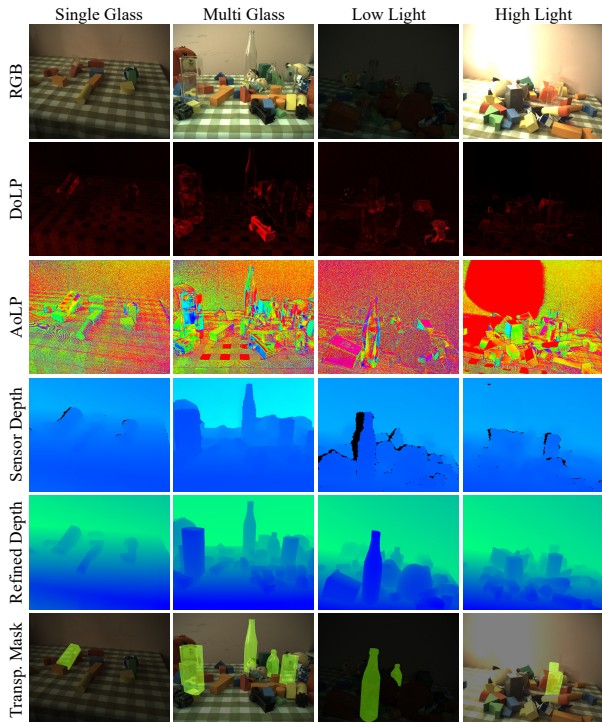

Figure 2. Example samples from the PTOD dataset.

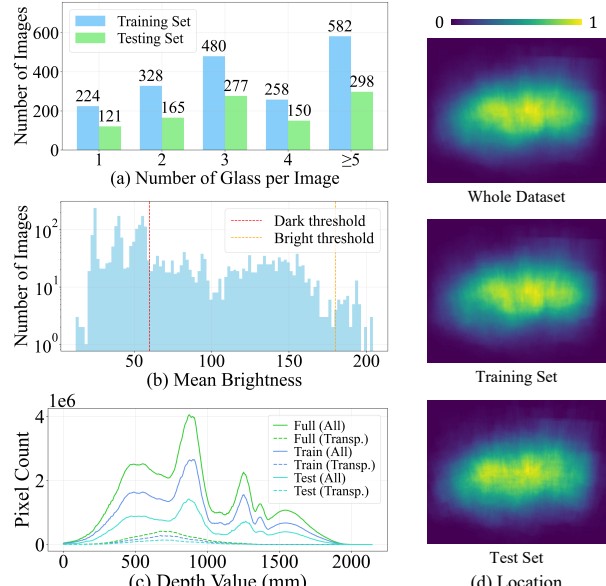

Figure 3. Statistics of the PTOD dataset. (a) Distribution of transparent object counts. (b) Image brightness distribution. (c) Pixel-wise depth value distribution. (d) Spatial distribution of transparent object locations.

larly to RGB inputs, extracting only general features without modeling their underlying physical significance. In contrast, our approach explicitly derives physically meaningful priors from polarization, including refractive index and surface orientation. These priors are used as geometric constraints to guide depth estimation, allowing accurate reconstruction of transparent object geometry even where RGB information is unreliable.

## 3. Polarized Transparent Object Dataset

We construct the **P**olarized **T**ransparent **O**bject **D**ataset (PTOD), a novel pixel-aligned RGB-polarization dataset for transparent object perception. PTOD is captured using a polarizer-array camera that records RGB images along with DoLP and AoLP maps at a resolution of $612 \times 512$. An active stereo depth sensor provides a metric depth map.

PTOD contains 2,878 scenes, each featuring at least one transparent glass object with diverse shapes and placements. To overcome transparency-induced failures in depth sensing, we adopt a replacement-based acquisition strategy that enables accurate geometry capture of transparent objects. High-quality ground-truth depth maps are obtained by refining predictions from Depth Anything v2 (Yang et al., 2024), using sensor depth as a metrically accurate reference. The full capture and refinement pipeline is detailed in Appendix B.

In addition to depth annotations, PTOD provides manually labeled segmentation masks for transparent regions, enabling joint learning of depth estimation and transparent object localization. As shown in Figures 2 and 3, PTOD exhibits substantial diversity in scene configurations, object geometries, and spatial distributions.

We note that HAMMER (Jung et al., 2023) is a related multimodal benchmark with RGB and polarization data. However, HAMMER provides instance-level masks that include opaque components, whereas our method requires masks that isolate transparent regions only to enforce polarization-derived physical priors, making it incompatible with our training protocol without substantial re-annotation.

## 4. Methodology

Accurate perception of transparent objects is critical for robotic manipulation, yet RGB-based methods struggle due to unreliable appearance cues. Polarization provides complementary geometric signals by encoding intrinsic surface properties that remain informative when appearance cues vanish. We introduce PolarDepth, which integrates an initial depth estimate from a foundational model with polarization-derived geometric constraints and selective refinement in transparent regions (Figure 4).

Given an input $\mathcal{I} = \{I_{rgb}, \rho, \phi\}$ containing RGB image, Degree of Linear Polarization (DoLP), and Angle of Linear Polarization (AoLP), PolarDepth refines depth by combining global and physics-based cues. A frozen foundational

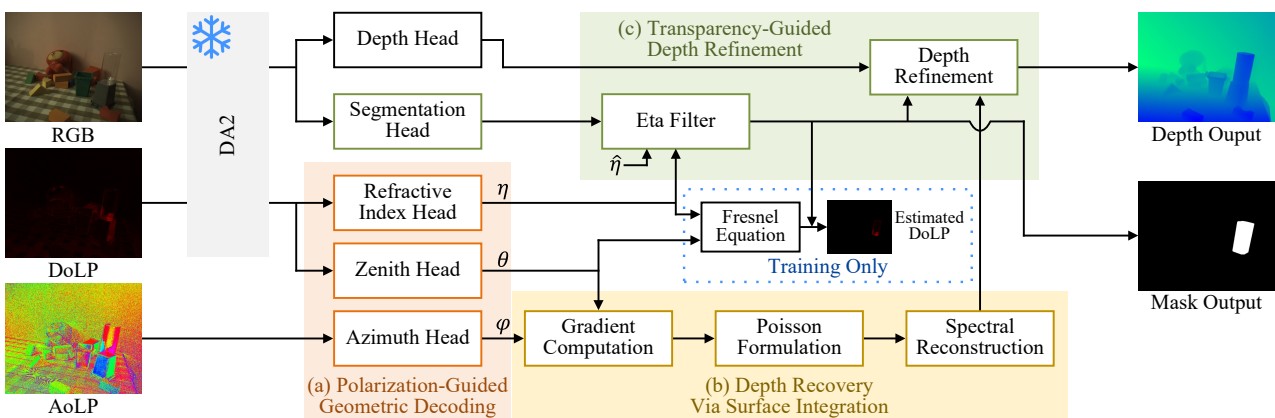

*Figure 4.* Overview of our Polar Depth Network. RGB and polarization inputs are processed by foundational depth model and task-specific heads to estimate global depth, polarization-derived geometric cues, and transparent object masks, which are then integrated to produce refined depth predictions for transparent scenes. The Fresnel Equation is enable only during training for supervise the prediction of refractive index ($\eta$) and zenith angle ($\theta$).

model predicts initial depth, while a polarimetric decoder estimates surface geometry from polarization measurements, which are selectively applied to transparent regions.

### 4.1. Polarization-Guided Geometric Decoding

Polarization measurements encode deterministic relationships between light reflection and surface geometry at dielectric interfaces. PolarDepth decodes physically meaningful parameters from polarization rather than treating them as auxiliary images. Under the Fresnel model, DoLP $\rho$ depends on the refractive index $\eta$ and zenith angle $\theta$, while AoLP $\phi$ corresponds to the azimuth angle $\varphi$ (see Appendix A). Specifically, for unpolarized incident light, the DoLP can be expressed as a nonlinear function of the Fresnel reflection coefficients:

$$\rho = \left| \frac{r_s^2 - r_p^2}{r_s^2 + r_p^2} \right|,$$
$$r_s = \frac{\cos\theta - \eta\cos\theta_t}{\cos\theta + \eta\cos\theta_t}, \ r_p = \frac{\eta\cos\theta - \cos\theta_t}{\eta\cos\theta + \cos\theta_t}, \quad (1)$$
$$\cos\theta_t = \sqrt{1 - \frac{1}{\eta^2}\sin^2\theta}.$$

AoLP is embedded as $\phi_e$ using a continuous embedding $\phi_e = [\sin(2\phi), \cos(2\phi)]$ to avoid angular discontinuities caused by the inherent $\pi$-periodicity (Lei et al., 2022), combined with DoLP to predict dense maps of $\eta$, $\theta$, and $\varphi$, which are then converted into surface normal field:

$$\mathbf{N} = [\sin\theta\cos\varphi, \sin\theta\sin\varphi, \cos\theta]^\top. \quad (2)$$

This process transforms raw polarization measurements into explicit geometric primitives that serve as the foundation for shape-constrained depth reconstruction.

### 4.2. Depth Recovery via Surface Integration

The inferred surface normals encode local geometry but lack global consistency. We recover a coherent depth map using a Poisson formulation that enforces gradient consistency in a least-squares sense. Given a predicted unit normal $\mathbf{N} = [n_x, n_y, n_z]^\top$, surface gradients are computed as:

$$p = -\frac{n_x}{n_z}, \ q = -\frac{n_y}{n_z}. \quad (3)$$

The depth map $Z$ is then recovered by solving a Poisson equation whose solution minimizes the discrepancy between the reconstructed and predicted gradients:

$$\nabla^2 Z = \text{div}(p, q) = \frac{\partial p}{\partial x} + \frac{\partial q}{\partial y}. \quad (4)$$

This is solved in the spectral domain via the Fast Fourier Transform (FFT):

$$\mathcal{F}(Z) = \frac{\mathcal{F}(\nabla \cdot (p, q))}{-4\pi^2(k_x^2 + k_y^2)}, \quad (5)$$

yielding a zero-mean depth map capturing detailed shape variations of transparent objects. The resulting $Z$ provides a relative, shape-constrained depth prior that complements the initial but globally consistent depth from foundational model. Implementation details are provided in Appendix C.

### 4.3. Transparency-Guided Depth Refinement

To guide selective application of polarization cues, PolarDepth predicts a transparency-aware segmentation using refractive index $\eta$ as a cue. A Gaussian attention highlights regions with typical refractive indices, enabling the network to distinguish transparent objects from opaque surfaces:

*Table 1.* Quantitative comparison against competing methods on the PTOD test set. We report depth estimation performance in transparent regions and over full scenes, as well as transparent object segmentation accuracy. Methods are grouped as monocular depth estimation, transparent object segmentation, and monocular multi-task frameworks. The Diffusion-based model is marked with ⋆, methods designed for transparent objects with †, and polarization-based methods with ◊. Underlined methods are evaluated using official pre-trained weights, as their training code is not available. For each metric, the best , second-best , and third-best performances are highlighted.

| Method | Depth (transparent) | | | | Depth | | | | Segmentation | | | |
|---|---|---|---|---|---|---|---|---|---|---|---|---|
| | $\delta_1 \uparrow$ | AbsRel↓ | RMSE↓ | SI-Log↓ | $\delta_1 \uparrow$ | AbsRel↓ | RMSE↓ | SI-Log↓ | IoU↑ | $F_\beta \uparrow$ | MAE↓ | BER↓ |
| Depth4ToM† | 0.416 | 0.423 | 0.289 | 0.500 | 0.375 | 0.444 | 0.374 | 1.195 | × | × | × | × |
| DA2 | 0.547 | 0.377 | 0.239 | 0.268 | 0.523 | 0.304 | 0.260 | 0.292 | × | × | × | × |
| Depth Pro | 0.354 | 0.562 | 0.411 | 0.357 | 0.468 | 0.404 | 0.248 | 0.249 | × | × | × | × |
| Marigold⋆ | 0.467 | 0.221 | 0.246 | 0.387 | 0.579 | 0.352 | 0.228 | 0.255 | × | × | × | × |
| DA3 | 0.303 | 0.290 | 0.232 | 0.266 | 0.500 | 0.255 | 0.221 | 0.261 | × | × | × | × |
| MoGe-2 | 0.565 | 0.242 | 0.367 | 0.238 | 0.551 | 0.230 | 0.295 | 0.217 | × | × | × | × |
| Trans4Trans† | × | × | × | × | × | × | × | × | 0.574 | 0.705 | 0.032 | 0.180 |
| PGSNet† ◊ | × | × | × | × | × | × | × | × | 0.882 | 0.936 | 0.010 | 0.038 |
| InvPT | 0.582 | 0.293 | 0.201 | 0.244 | 0.575 | 0.307 | 0.240 | 0.264 | 0.591 | 0.729 | 0.029 | 0.182 |
| TaskPrompter | 0.537 | 0.382 | 0.268 | 0.319 | 0.546 | 0.287 | 0.254 | 0.292 | 0.573 | 0.691 | 0.032 | 0.187 |
| GW-Depth† | 0.614 | 0.246 | 0.193 | 0.284 | 0.597 | 0.283 | 0.244 | 0.285 | 0.879 | 0.934 | 0.016 | 0.042 |
| MODEST† | 0.735 | 0.252 | 0.144 | 0.186 | 0.605 | 0.251 | 0.207 | 0.252 | 0.544 | 0.643 | 0.029 | 0.217 |
| **PolarDepth**† ◊ **(Ours)** | 0.859 | 0.142 | 0.108 | 0.134 | 0.612 | 0.222 | 0.200 | 0.209 | 0.893 | 0.943 | 0.008 | 0.032 |

$$\omega(\eta) = \exp\left(-\frac{(\eta - \hat{\eta})^2}{2\hat{\sigma}^2}\right), \qquad (6)$$

where $\hat{\eta}$ and $\hat{\sigma}$ are learnable parameters representing the mean refractive index and its bandwidth, respectively. A segmentation head aggregates multi-scale features from the backbone to produce initial segmentation. Subsequently, the refractive-aware attention mechanism leverages the predicted $\eta$ as a gating signal to highlight dielectric regions.

Depth refinement is performed in an iterative manner. $Z$ is gated by the transparency mask and fused with the foundational depth prediction, injecting local geometric details only in transparent regions. Three residual convolutions are then adopted to ensures global consistency and smooth transitions between transparent and opaque areas. This selective fusion improves geometric fidelity without artifacts.

### 4.4. Joint Physical and Semantic Supervision

To ensure physically consistent learning, PolarDepth is trained under a joint supervision framework coupling depth, segmentation, and polarization-derived physical consistency:

$$\mathcal{L}_{total} = \lambda_{dep}\mathcal{L}_{dep} + \lambda_{seg}\mathcal{L}_{seg} + \lambda_{phys}\mathcal{L}_{phys}. \quad (7)$$

Depth supervision combines scale-invariant logarithmic loss, multi-scale gradient loss, and edge-aware smoothness. Segmentation uses BCE, IoU, and boundary-aware losses. Physical consistency is enforced by comparing the predicted DoLP (i.e., $\hat{\rho}$), computed via the Fresnel model (Equation 1) with input polarization, ensuring physically plausible predictions. Near the Brewster angle, we apply a weak zenith

regularization to favor physically plausible surface orientations:

$$L_{phys} = \|\rho - \hat{\rho}\|_1 + 0.01 \cdot \frac{1}{n}\sum_{i=1}^{n} \theta_i^2 \qquad (8)$$

Together, these components enable accurate, semantically meaningful, and physically grounded depth reconstruction for transparent objects.

## 5. Experiments

### 5.1. Experimental Setting

**Implementation Details.** PolarDepth adopts Depth Anything v2 (Yang et al., 2024) with a ViT-S backbone as the foundational depth model, with frozen pre-trained weights. Task-specific heads for transparent region segmentation, refractive index estimation, surface orientation, and depth refinement are implemented using lightweight convolutional layers and initialized randomly. The model is trained using AdamW ($lr = 1 \times 10^{-4}$, $\beta = (0.9, 0.999)$, weight decay $= 1 \times 10^{-2}$). Training is conducted for 100 epochs on two NVIDIA L40 GPUs with a batch size of 48. All competing methods are trained or fine-tuned on the PTOD training split if the code is available, and evaluated on the test set.

**Evaluation Metrics.** Depth is evaluated using Delta 1 ($\delta_1$), mean absolute relative error (AbsRel), root mean square error (RMSE), and scale-invariant logarithmic error (SI-Log). Segmentation is assessed using intersection over union (IoU), F-measure ($F_\beta$), mean absolute error (MAE), and balanced error rate (BER). Higher values indicate better performance (↑) for $\delta_1$, IoU, and $F_\beta$, while lower values are preferable (↓) for AbsRel, RMSE, SI-Log, MAE, and BER.

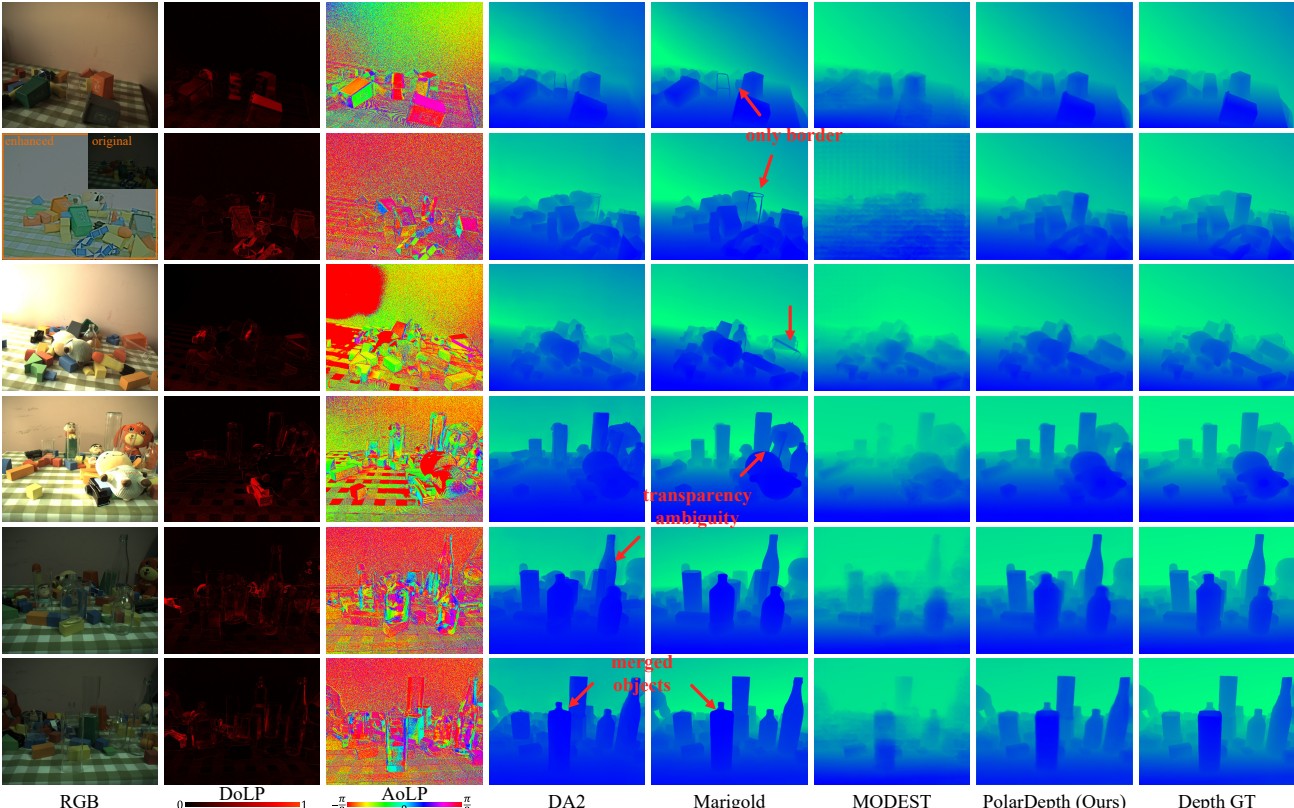

| RGB | DoLP | AoLP | DA2 | Marigold | MODEST | PolarDepth (Ours) | Depth GT |

*Figure 5.* Qualitative comparison of PolarDepth against state-of-the-art depth estimation methods. PolarDepth successfully recovers coherent geometry in textureless transparent regions (Rows 1-3) and resolves depth ambiguities in complex overlapping scenarios (Rows 4-5). By leveraging polarization-based physical priors, our framework accurately captures spatial variations and distinguishes distinct objects (Rows 6–7) that other methods collapse into single planes under RGB-based estimation.

### 5.2. Comparison to Prior Works

We compare PolarDepth against 11 state-of-the-art approaches, including 5 monocular depth estimation methods, 2 transparent object segmentation methods, and 4 monocular multi-task frameworks. Among monocular depth estimators, Depth4ToM targets transparent and reflective objects, DA2 (Yang et al., 2024), Depth Pro (Bochkovskiy et al., 2024), Marigold (Ke et al., 2024), DA3 (Lin et al., 2025), and MoGe-2 (Wang et al., 2026) are general-purpose depth models. Depth Pro emphasizes high-frequency details reconstruction, while Marigold leverages rich priors from the diffusion model. PGSNet (Mei et al., 2022) and Trans4Trans (Zhang et al., 2021) perform transparent object segmentation, with only PGSNet exploiting polarization. Multi-task methods Invpt (Ye & Xu, 2022a), Taskprompter (Ye & Xu, 2022b), GW-Depth (Liang et al., 2023), and MODEST (Liu et al., 2025) predict both depth and segmentation; GW-Depth additionally relies on transparent-region edge supervision.

Table 1 shows that PolarDepth achieves the best performance across all depth metrics, with particularly large gains in transparent regions. RGB-reliant methods like GW-Depth

and MODEST benefit from joint tasks, their performance remains limited by unreliable RGB appearance cues in transparent regions. In contrast, PolarDepth effectively overcomes this ambiguity by incorporating polarization-derived physical constraints. For segmentation, PolarDepth and PGSNet outperform RGB-based methods, highlighting the critical role of polarization cues for transparent object perception. The inferior performance of other methods can be attributed to their general-purpose design and reliance on RGB appearance, which is inherently unstable for transparent objects.

Figure 5 illustrates representative qualitative results:

1. In the first three rows, transparent regions exhibit textures similar to the background due to light transmission under different light conditions. Only PolarDepth successfully recovers the correct geometry by leveraging polarization cues that reveal continuous and coherent object shape, whereas other methods mainly infer object boundaries from weak refraction cues near edges.

2. The 4th and 5th rows present more complex scenes involving overlapping transparent objects or interactions with other objects. Competing methods are confused by

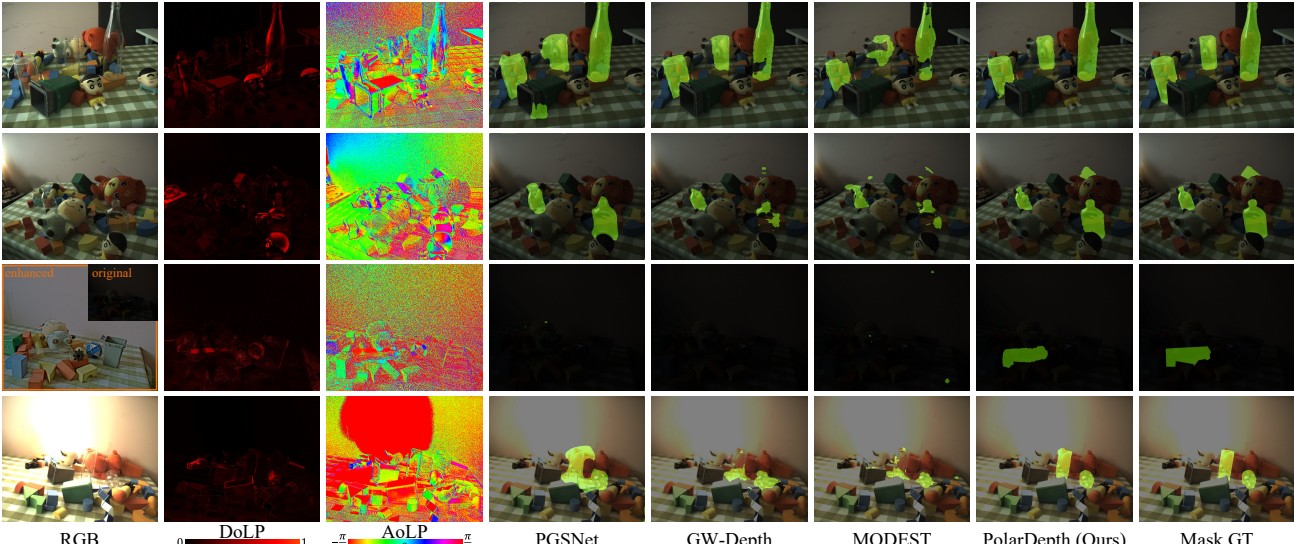

RGB    DoLP    AoLP    PGSNet    GW-Depth    MODEST    PolarDepth (Ours)    Mask GT

*Figure 6.* Qualitative comparison of PolarDepth against state-of-the-art segmentation methods. PolarDepth utilizes polarimetric information to achieve precise semantic localization and boundary recovery. Our method consistently outperforms competing approaches that are easily confused by transmitted background textures and the lack of reliable visual edges.

multi-level transparency-induced ambiguity. PolarDepth successfully separates and estimates their depths by integrating polarization-derived geometry constraints.

3. In the last two rows, competing methods tend to assign similar depths to distinct transparent objects because RGB appearance provides insufficient distance information. Competing methods collapse them into a single plane. In contrast, PolarDepth exploits polarization-based physical priors that encode spatial shape variation, enabling accurate depth estimation.

### 5.3. Illumination Robustness

PolarDepth leverages the inherent robustness of polarimetric cues to varying illumination. According to the Fresnel reflection model, DoLP and AoLP are determined primarily by surface geometry and the refractive index. While the absolute intensity of reflected light varies significantly between localized point sources and diffuse environment illumination, the polarization state, defined by the orientation and ratio of electric field components, remains theoretically coupled to the surface normal. Our polarization-guided geometric decoding explicitly exploits this illumination-invariant property, enabling consistent geometry recovery across diverse environments.

The PTOD dataset encompasses various real-world lighting scenarios (Figures 2 and 3), which we categorize into bright natural, dim natural, and artificial illumination for a sub-group analysis. As shown in Table 2 (reported as Transparent Regions / Full Image), PolarDepth maintains stable

performance across all conditions, with significant gains in transparent regions. These results confirm that polarization-derived physical priors provide a reliable geometric signal that is substantially less sensitive to intensity fluctuations than purely RGB-based appearance cues.

### 5.4. Ablation Study

We conduct extensive ablation experiments to assess the contribution of each component within PolarDepth (Table 3). Specifically, we examine the impact of the refractive-aware segmentation branch, the physics-grounded geometry decoding, and the iterative refinement stages. The baseline ($B$) uses RGB input only, and corresponds to original Depth Anything V2. The results demonstrate that each module is indispensable, collectively contributing to the framework's state-of-the-art accuracy and geometric stability.

**Importance of Segmentation Branch.** The segmentation branch serves as a crucial spatial gate for polarimetric information. Since the Fresnel equations vary across material types, applying polarization constraints without semantic guidance ($C$) leads to significant performance degradation ($\delta_1$ from 0.612 to 0.552). As illustrated in Figure 7, the lack of spatial gating causes the injection of incorrect geometric priors into non-transparent regions, resulting in severe depth artifacts. Furthermore, removing the Eta Filter ($D$) reduces the network's ability to localize dielectric boundaries based on refractive index, leading to a drop in segmentation metrics.

**Importance of Physical Constraint.** We evaluate the neces-

*Table 2.* Quantitative comparison under varying illumination conditions. Metrics are reported as (Transparent Regions / Full Image). The stable performance of PolarDepth across all lighting scenarios validates that polarization-derived physical priors provide a robust, illumination-invariant geometric anchor, outperforming RGB-based methods.

| Condition / Method | $\delta_1 \uparrow$ | AbsRel $\downarrow$ | RMSE $\downarrow$ | SI-Log $\downarrow$ |
|---|---|---|---|---|
| (1) Bright Natural Illumination | | | | |
| DA2 | 0.551 / 0.529 | 0.361 / 0.291 | 0.228 / 0.247 | 0.260 / 0.281 |
| Marigold | 0.493 / 0.592 | 0.194 / 0.324 | 0.232 / 0.225 | 0.293 / 0.244 |
| GW-Depth | 0.626 / 0.599 | 0.235 / 0.284 | 0.187 / 0.232 | 0.275 / 0.273 |
| MODEST | 0.759 / 0.607 | 0.242 / 0.245 | 0.134 / 0.203 | 0.181 / 0.247 |
| PolarDepth (Ours) | 0.854 / 0.612 | 0.141 / 0.218 | 0.112 / 0.196 | 0.134 / 0.207 |
| (2) Dim Natural Illumination | | | | |
| DA2 | 0.523 / 0.507 | 0.398 / 0.312 | 0.254 / 0.278 | 0.273 / 0.305 |
| Marigold | 0.411 / 0.531 | 0.289 / 0.385 | 0.262 / 0.235 | 0.540 / 0.276 |
| GW-Depth | 0.571 / 0.582 | 0.265 / 0.274 | 0.203 / 0.266 | 0.288 / 0.311 |
| MODEST | 0.663 / 0.596 | 0.261 / 0.257 | 0.157 / 0.212 | 0.190 / 0.257 |
| PolarDepth (Ours) | 0.858 / 0.613 | 0.125 / 0.220 | 0.096 / 0.201 | 0.131 / 0.206 |
| (3) Artificial Illumination | | | | |
| DA2 | 0.542 / 0.501 | 0.379 / 0.319 | 0.240 / 0.262 | 0.273 / 0.294 |
| Marigold | 0.442 / 0.578 | 0.197 / 0.376 | 0.255 / 0.216 | 0.446 / 0.247 |
| GW-Depth | 0.608 / 0.583 | 0.241 / 0.279 | 0.188 / 0.238 | 0.293 / 0.271 |
| MODEST | 0.726 / 0.581 | 0.258 / 0.249 | 0.150 / 0.202 | 0.187 / 0.248 |
| PolarDepth (Ours) | 0.834 / 0.583 | 0.163 / 0.227 | 0.107 / 0.201 | 0.132 / 0.209 |

*Table 3.* Quantitative ablation analysis. The metrics demonstrate that removing the segmentation branch, physical priors, or refinement stages leads to significant degradation in depth accuracy and geometric stability.

| Model Ablation | Depth (transparent) | | | | Depth | | | | Segmentation | | | |
|---|---|---|---|---|---|---|---|---|---|---|---|---|
| | $\delta_1 \uparrow$ | AbsRel $\downarrow$ | RMSE $\downarrow$ | SI-Log $\downarrow$ | $\delta_1 \uparrow$ | AbsRel $\downarrow$ | RMSE $\downarrow$ | SI-Llog $\downarrow$ | IoU $\uparrow$ | $F_\beta \uparrow$ | MAE $\downarrow$ | BER $\downarrow$ |
| **A PolarDepth (original)** | **0.859** | **0.142** | **0.108** | **0.134** | **0.612** | **0.222** | **0.200** | **0.209** | **0.893** | **0.943** | **0.008** | **0.032** |
| B Input RGB only | 0.547 | 0.377 | 0.239 | 0.268 | 0.523 | 0.304 | 0.260 | 0.292 | $\times$ | $\times$ | $\times$ | $\times$ |
| C A *w/o* segmentation branch | 0.688 | 0.235 | 0.160 | 0.199 | 0.552 | 0.636 | 0.869 | 0.512 | $\times$ | $\times$ | $\times$ | $\times$ |
| D A *w/o* Eta Filter | 0.816 | 0.160 | 0.119 | 0.141 | 0.609 | 0.222 | 0.200 | 0.209 | 0.854 | 0.905 | 0.011 | 0.052 |
| E A *w/o* depth recovery | 0.668 | 0.242 | 0.167 | 0.230 | 0.578 | 0.275 | 0.212 | 0.274 | 0.886 | 0.938 | 0.008 | 0.037 |
| F A *w/o* surface integration | 0.686 | 0.181 | 0.141 | 0.159 | 0.581 | 0.263 | 0.227 | 0.248 | 0.868 | 0.927 | 0.010 | 0.044 |
| G Training *w/o* $\mathcal{L}_{phys}$ | 0.753 | 0.169 | 0.132 | 0.156 | 0.605 | 0.215 | 0.193 | 0.200 | 0.855 | 0.920 | 0.011 | 0.044 |
| H A *w/o* Depth Refinement | 0.787 | 0.174 | 0.128 | 0.160 | 0.602 | 0.258 | 0.215 | 0.211 | 0.884 | 0.937 | 0.008 | 0.038 |
| I H *w/* Depth Refiner $\times 2$ | 0.855 | 0.144 | 0.109 | 0.134 | 0.605 | 0.234 | 0.209 | 0.212 | 0.891 | 0.939 | 0.008 | 0.032 |
| J H *w/* Depth Refiner $\times 4$ | 0.854 | 0.143 | 0.114 | 0.129 | 0.615 | 0.217 | 0.202 | 0.208 | 0.889 | 0.933 | 0.008 | 0.033 |

sity of physics-based modeling by replacing the geometric decoding and surface integration with direct CNN regression (E). This reduces $\delta_1$ in transparent regions from 0.859 to 0.668, confirming that treating polarization merely as an appearance cue underutilizes its inherent geometric information (Figure 8). Removing the Poisson integration while keeping the decoding head (F) also yields inferior results, as it lacks the global spatial consistency provided by the spectral-domain optimization. Finally, training without the physical consistency loss $\mathcal{L}_{phys}$ (G) leads to unstable geometry, confirming the role of Fresnel supervision in grounding the network.

**Importance of Depth Refinement.** We assess the impact of the iterative refinement process on metric accuracy. Without the refinement module (H), the predicted transparent objects often exhibit incorrect global scales and appear spatially disconnected from the scene context (Figure 9). Comparisons between varying amounts of refinement stages (I-J) indicate

that an iterative approach improves both global consistency and local sharpness. Three refinement stages provide an optimal balance between accuracy and computation, effectively reconciling the relative polarimetric shapes with the metric foundational base.

## 5.5. Model Efficiency

Table 4 summarizes model complexity and runtime. Thanks to the robust inference capabilities of existing monocular depth estimation models, we were able to overcome the long-standing challenge of dealing with transparent glass objects by effectively utilizing polarization cues. PolarDepth introduces minimal parameter and moderate computational overhead, as most polarization-related components are lightweight or non-learnable. This efficiency makes PolarDepth well suited for real-world robotic applications, where accurate perception of transparent objects is required under limited computational resources.

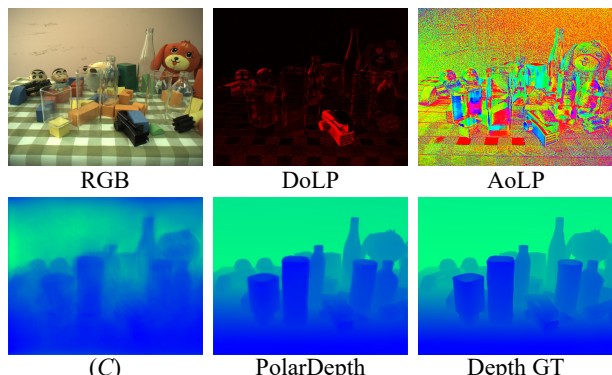

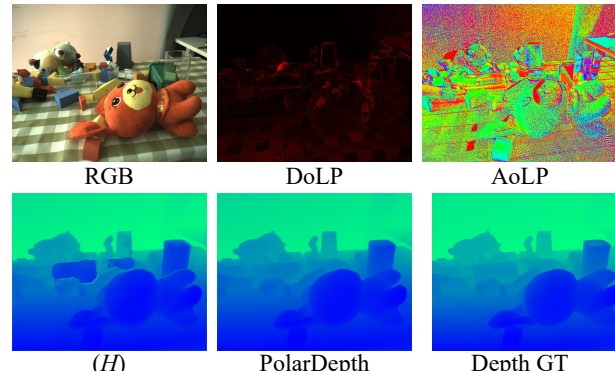

*Figure 7.* Impact of semantic gating. Without the segmentation branch, polarimetric geometric priors are incorrectly applied to the entire scene, introducing significant artifacts in non-transparent background regions.

*Figure 9.* Ablation of iterative refinement. The iterative refinement process ensures that transparent objects are correctly scaled and integrated into the global metric layout of the scene.

*Table 4.* Quantitative analysis of model parameters, FLOPs, and inference latency. PolarDepth introduces negligible parameter overhead relative to the ViT-S backbone and maintains competitive runtime performance. These results highlight the efficiency of our polarization-guided refinement compared to larger foundational or optimization-based approaches.

| Method | Backbone | Params (M) | Flops (G) | Infer. time (ms) |
|---|---|---|---|---|
| Depth4ToM | ViT-L | 341 | 295 | 33.9 |
| DA2 | ViT-S | 24 | 41 | 23.0 |
| Depthpro | ViT-L | 504 | 4370 | 724.7 |
| Marigold | StableDiff. | 949 | - | 311.5 |
| DA3 | ViT-L | 358 | 121 | 39.6 |
| MoGe-2 | ViT-S | 35 | - | 30.1 |
| Trans4Trans | PVT-M | 44 | 121 | 56.2 |
| PGSNet | Conformer-B | 303 | 291 | 86.1 |
| InvPT | ViT-L | 423 | 669 | 209.6 |
| TaskPropmter | ViT-L | 401 | 497 | 143.2 |
| GW Depth | DETR-Res50 | 61 | 506 | 53.6 |
| MODEST | ViT-B | 102 | 289 | 31.6 |
| PolarDepth (Ours) | ViT-S | 24 | 118 | 25.1 |

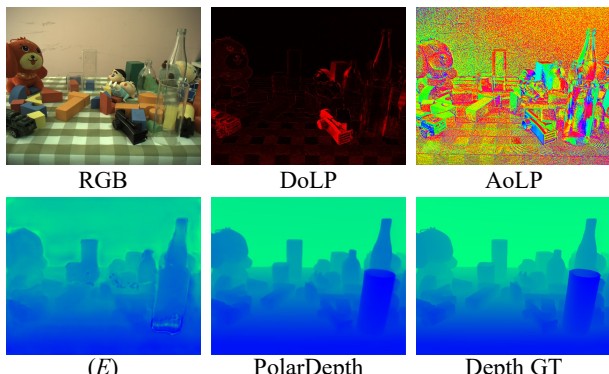

*Figure 8.* Physical decoding *vs.* direct regression. Removing the physics-based Poisson integration fails to capture the intricate curvatures of transparent surfaces, resulting in inaccurate geometry.

## 6. Conclusion

This work introduces PolarDepth, a monocular depth estimation framework for transparent objects guided by physically grounded polarization priors. By modeling the relationship between polarization and surface geometry at dielectric interfaces, PolarDepth recovers reliable depth in transparent regions where RGB cues are ambiguous. The framework integrates initial RGB-based depth with polarization-derived geometric constraints and selectively refines predictions using corresponding segmentation. Experiments on the PTOD dataset show state-of-the-art performance in depth estimation and transparent object segmentation with real-time inference, demonstrating the effectiveness of polarization as a complementary modality for transparent object perception.

A current limitation, common to polarization-based methods, is sensitivity to surface material properties that weakly polarize reflected light. Future work will explore robustness across a broader range of transparent materials and improved modeling of low-polarization regions.

## Impact Statement

This paper presents work whose goal is to advance the field of machine learning and computer vision by improving depth perception for transparent objects using polarization cues. The proposed methods are intended to support reliable perception in applications such as robotics, augmented reality, and scene understanding. We do not identify any specific ethical or societal concerns beyond those commonly associated with perception technologies, and no issues require explicit discussion at this time.

**Acknowledgements.** This work is supported in part by the National Key Research and Development Program of China (No. 2022ZD0210500), the National Natural Science Foundation of China (Grant No. 62502068), the Guangdong Natural Science Funds for Distinguished Young Scholars (Grant 2023B1515020097), the Singapore Ministry of Education Academic Research Fund Tier 1 (Proposal ID: 24-SIS-SMU-015), and the Lee Kong Chian Fellowships.

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

# A. Polarization Fundamentals

This appendix provides background on polarization imaging to support the main paper. We review (1) the definition and representation of polarization, (2) its physical relationship with surface orientation, and (3) the acquisition of polarization images using a polarizer-array camera. This material complements Section 4 and is included for completeness.

### A.1. What Is Polarization and How Is It Represented?

Light is an electromagnetic wave whose electric field oscillates perpendicular to the direction of propagation. Polarization describes the orientation and temporal behavior of this oscillation. While natural illumination is typically unpolarized, interactions with surfaces (e.g., reflection and refraction) induce partial polarization that encodes geometric information.

A common and compact representation of polarization is given by the Stokes vector:

$$S = [S_0, S_1, S_2, S_3]^\top, \tag{9}$$

where $S_0$ denotes total intensity, $S_1$ and $S_2$ describe linear polarization, and $S_3$ corresponds to circular polarization. Since reflections from dielectric surfaces primarily induce linear polarization, we focus on $S_0$, $S_1$, and $S_2$.

From the Stokes parameters, two scalar quantities are commonly derived:

- Degree of Linear Polarization (DoLP) $\rho$, measuring polarization strength,
- Angle of Linear Polarization (AoLP) $\phi$, describing the dominant orientation of the electric field.

They are computed as:

$$\rho = \frac{\sqrt{S_1^2 + S_2^2}}{S_0}, \phi = \frac{1}{2} arctan(\frac{S_2}{S_1}) \tag{10}$$

These two quantities form the polarization representation $(\rho, \phi)$ used throughout this work and serve as the primary input to our polarization-guided geometric decoding module (Section 4).

### A.2. Relationship Between Polarization and Surface Orientation

The polarization state of reflected light is governed by the Fresnel reflection model, which establishes a deterministic relationship between polarization and surface geometry at dielectric interfaces.

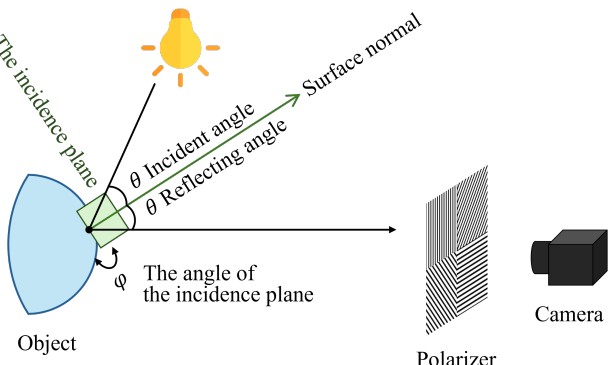

*Figure 10.* Polarization formation and geometric interpretation. Unpolarized incident light becomes partially polarized upon reflection from a transparent surface. The reflected light lies in the plane of incidence defined by the surface normal and the viewing direction, with equal incident and reflection angles $\theta$. A polarization camera captures the reflected light, enabling estimation of the reflecting angle $\theta$ and the plane-of-incidence orientation $\phi$, from which the surface orientation relative to the viewer can be determined.

As illustrated in Figure 10, natural illumination is typically unpolarized and becomes partially polarized after reflecting from a surface. For transparent objects, surface interfaces induce limited diffuse reflection or absorption, and the reflected light follows specular reflection with equal incident and reflection angles $\theta$. The reflected ray, surface normal, and viewing direction lie on the same plane of incidence, whose orientation is characterized by the angle $\phi$ (Miyazaki et al., 2004).

For unpolarized incident light, Fresnel reflection causes different attenuation of the electric field components parallel and perpendicular to the plane of incidence. As a result:

- The DoLP $\rho$ depends on the zenith angle $\theta$ and the refractive index $\eta$,
- The AoLP $\phi$ aligns with the azimuth angle $\varphi$ of the surface normal (up to a $\pi$ ambiguity).

This physically grounded relationship enables polarization measurements to encode surface orientation even in regions where RGB appearance cues are unreliable, such as transparent or textureless objects. Unlike learning-based monocular cues, this mapping is deterministic and scene-independent.

In the main paper (Section 4), we exploit this property by explicitly predicting $\eta$, $\theta$, and $\varphi$ from polarization measurements and converting them into geometry cues, rather than treating polarization as a generic auxiliary modality and instead leverages its intrinsic physical meaning.

### A.3. Fresnel Model for Transparent Glass Surfaces

For transparent dielectric materials such as glass, the polarization state of reflected light is governed by Fresnel reflection. Unlike opaque surfaces, glass exhibits minimal diffuse reflection; instead, polarization arises primarily from specular reflection at the air–glass interface. This makes Fresnel theory particularly well suited for modeling polarization cues on transparent objects.

Under the Fresnel model, the reflection coefficients for electric field components perpendicular ($r_s$) and parallel ($r_p$) to the plane of incidence are given by:

$$r_s = \frac{\cos\theta - \eta\cos\theta_t}{\cos\theta + \eta\cos\theta_t}, \ r_p = \frac{\eta\cos\theta - \cos\theta_t}{\eta\cos\theta + \cos\theta_t}, \tag{11}$$

where $\theta$ is the zenith angle of the surface normal relative to the viewing direction, $\eta$ denotes the refractive index of the material, and $\theta_t$ is the transmission angle satisfying Snell's law:

$$\cos\theta_t = \sqrt{1 - \frac{1}{\eta^2}\sin^2\theta}. \tag{12}$$

For unpolarized incident light, the Degree of Linear Polarization (DoLP) of the reflected light is determined by the imbalance between $r_s$ and $r_p$:

$$\rho = \left|\frac{r_s^2 - r_p^2}{r_s^2 + r_p^2}\right|. \tag{13}$$

This expression reveals that DoLP depends deterministically on both the surface zenith angle $\theta$ and the refractive index $\eta$. Meanwhile, the Angle of Linear Polarization (AoLP) aligns with the orientation of the plane of incidence, which corresponds to the azimuth angle $\varphi$ of the surface normal up to a $\pi$ ambiguity.

These properties are especially important for transparent glass objects: while RGB appearance may be dominated by background transmission or refraction, polarization remains directly linked to surface geometry through Fresnel reflection. This deterministic and material-aware relationship motivates our design choice to explicitly decode $\eta$, $\theta$, and $\varphi$ from polarization measurements in PolarDepth, rather than treating polarization as a generic visual modality.

### A.4. Polarization Image Acquisition

The Polarized Transparent Object Dataset (PTOD) is captured using a polarizer-array camera (FLIR Blackfly-S) that simultaneously records four fixed linear polarization orientations ($0°$, $45°$, $90°$, and $135°$). For each $2 \times 2$ superpixel, the camera measurs four intensity values:

$$I_0, I_{45}, I_{90}, I_{135}. \tag{14}$$

These measurements are used to compute the linear Stokes parameters:

$$S_0 = I_0 + I_{90}, \ S_1 = I_0 - I_{90}, \ S_2 = I_{45} - I_{135}. \tag{15}$$

From these, DoLP $\rho$ and AoLP $\phi$ are obtained using Equation 10. The resulting polarization images are spatially aligned with the RGB image and serve as direct inputs to PolarDepth, enabling single-shot acquisition suitable for real-world robotic perception.

## B. Construction of the PTOD Dataset

This appendix details the construction pipeline of the Polarized Transparent Object Dataset (PTOD), which provides pixel-aligned RGB, polarization, depth, and segmentation annotations for transparent objects. The dataset design overcomes the fundamental challenges of sensing transparent surfaces and supports the training and evaluation of polarization-guided depth estimation methods.

### B.1. Motivation and Data Acquisition Strategy

Accurate depth sensing of transparent objects is fundamentally unreliable due to light transmission, refraction, and multi-path interference, which violate the assumptions of most active depth sensors. As a result, directly capturing ground-truth depth using commodity depth sensors often produces missing, distorted, or biased measurements.

To address this challenge, we employ a replacement-based capture strategy. For each scene, we first place transparent objects and record an RGB image along with corresponding DoLP and AoLP maps using a polarizer-array camera (FLIR Blackfly-S). Next, without changing the camera pose or scene configuration, the transparent object is replaced by an opaque replica with identical geometry and carefully aligned pose. An active stereo depth sensor (Intel RealSense D435i) then captures a depth map, which serves as a proxy for the true geometry.

This strategy decouples optical complexity from geometric accuracy, yielding reliable depth supervision while preserving the original polarization and appearance cues of transparent materials.

### B.2. Dense Depth Alignment and Refinement

Although the sensor depth provides *metrically accurate geometry*, it is typically *sparse, noisy, and low-resolution*. In contrast, learning-based depth predictions are *dense, smooth and clear* but often suffer from *scale ambiguity and global bias*.

To combine their complementary strengths, **gradient magnitude maps** are first computed using Sobel operators to emphasizes structural boundaries. An **Enhanced Correlation Coefficient (ECC)** optimizer estimates a homography to align the gradient maps, correcting spatial misalignment. After alignment, **Random Sample Consensus (RANSAC)** robustly estimates a linear scale-and-shift transformation, resolving scale ambiguity while rejecting outliers.

Applying the transformation yields a dense, metrically accurate depth map that is pixel-aligned with the RGB and polarization images. This refined depth serves as the final ground-truth annotation used in PTOD.

### B.3. Transparent Object Mask Annotations

PTOD also includes manually annotated pixel-level segmentation masks for transparent objects. These annotations support transparent object localization and enable joint learning of segmentation and depth estimation, which is critical for downstream tasks such as robotic grasping and manipulation.

## C. Poisson-Based Surface Integration for Depth Recovery

This appendix provides implementation details for the Poisson-based surface integration used in the Depth Recovery via Surface Integration subsection of the main paper.

### C.1. From Surface Normals to Depth Gradients

Given a predicted surface normal field $N = [n_x, n_y.n_z]^\top$, the depth gradients along the horizontal and vertical directions are computed as:

$$p = \frac{\partial Z}{\partial x} = -\frac{n_x}{n_z}, \; q = \frac{\partial Z}{\partial y} = -\frac{n_y}{n_z}. \tag{16}$$

To ensure numerical stability, especially in regions where the surface normal is close to horizontal, we impose $n_z \leftarrow sign(n_z) \cdot \max(|n_z|, \epsilon)$ with $\epsilon = 10^{-3}$. This treatment avoids gradient explosion while preserving geometric fidelity.

## C.2. Poisson Formulation of Surface Integration

The depth map $Z$ is recovered by enforcing consistency between its gradients and the predicted gradient field $(p, q)$. This is formulated as a Poisson equation:

$$\nabla^2 Z = \nabla \cdot (p, q), \tag{17}$$

where the divergence is given by:

$$\nabla \cdot (p, q) = \frac{\partial p}{\partial x} + \frac{\partial q}{\partial y}. \tag{18}$$

We discretize the divergence using finite differences, applying central differences for interior pixels and forward or backward differences along image boundaries. This ensures stable handling of edge pixels without imposing artificial padding or boundary assumptions.

## C.3. Spectral Solution via Fast Fourier Transform

The Poisson equation is solved efficiently in the frequency domain. Let $f = \nabla \cdot (p, q)$. Taking the 2D Fourier transform yields:

$$\mathcal{F}(Z) = \frac{\mathcal{F}(f)}{-4\pi^2(k_x^2 + k_y^2)}, \tag{19}$$

where $k_x$ and $k_y$ denote the spatial frequencies along the horizontal and vertical axes, respectively.

To avoid division by zero at the zero-frequency (DC) component, the denominator is clamped to a non-zero value at $(k_x, k_y) = (0, 0)$, and the corresponding Fourier coefficient of $Z$ is explicitly set to zero:

$$\mathcal{F}(Z)_{(0,0)} = 0. \tag{20}$$

This enforces a zero-mean depth solution, which removes the global depth offset and preserves only relative shape information. Since the integration result is later fused with a globally consistent depth prediction, this offset does not affect the final reconstruction.

The final depth map is obtained by applying the inverse Fourier transform:

$$Z = \mathcal{F}^{-1}(\mathcal{F}(Z)) \tag{21}$$

## C.4. Output Normalization and Relation to the Main Method

The depth map is centered by subtracting its spatial mean:

$$Z \leftarrow Z - \frac{1}{HW}\Sigma_{x,y}Z(x, y), \tag{22}$$

yielding a relative, shape-constrained depth prior. This Poisson-based integration corresponds to Depth Recovery module in Section 4. The resulting depth map serves as a local geometric prior, which is selectively fused with the global depth prediction under segmentation guidance.

The entire procedure is fully differentiable and implemented using FFT operations, enabling efficient batch processing and seamless integration into the training and inference pipeline.

# D. Additional Experimental Analysis

This section provides additional experimental results and comparative analysis to support the findings presented in the main paper. We evaluate: (1) the effectiveness of physically grounded geometric decoding compared to naive feature fusion baselines, and (2) the cross-dataset generalization capability of our framework on the HAMMER dataset (Jung et al., 2023).

## D.1. Naive Feature Fusion *vs.* Geometric Decoding

To verify that the performance gains of PolarDepth stem from our physically grounded geometric decoding rather than the mere addition of polarization as a generic visual feature, we conduct a fairness study. We implement a "Naive Fusion" baseline (following the style of DPS-Net (Tian et al., 2023)) by concatenating polarization features (DoLP and AoLP) into the backbones of state-of-the-art monocular models (e.g., DA2 (Yang et al., 2024) and MODEST (Liu et al., 2025)).

As shown in Table 5, naively concatenating polarization features yields only marginal improvements in $\delta_1$ and AbsRel. This confirms that generic feature fusion is insufficient for handling the complex refraction and transmission effects inherent to transparent objects. In contrast, PolarDepth's superior performance demonstrates that our success stems from physically grounded decoding rather than the simple inclusion of an additional modality.

*Table 5.* Comparison of naive feature fusion vs. the proposed PolarDepth framework. Metrics are reported as (Transparent Regions / Full Image).

| Method | Fusion Mode | $\delta_1 \uparrow$ | AbsRel $\downarrow$ | RMSE $\downarrow$ | SI-Log $\downarrow$ |
|---|---|---|---|---|---|
| DA2 | RGB only | 0.547 / 0.523 | 0.377 / 0.304 | 0.239 / 0.260 | 0.268 / 0.292 |
| DA2 | Concatenation | 0.556 / 0.544 | 0.303 / 0.276 | 0.219 / 0.251 | 0.262 / 0.277 |
| MODEST | RGB only | 0.735 / 0.605 | 0.252 / 0.251 | 0.144 / 0.207 | 0.186 / 0.252 |
| MODEST | Concatenation | 0.723 / 0.607 | 0.254 / 0.242 | 0.143 / 0.206 | 0.186 / 0.246 |

## D.2. Cross-Dataset Generalization on HAMMER

We further evaluate the generalization capability of PolarDepth on the HAMMER dataset (Jung et al., 2023). Since HAMMER exhibits high frame redundancy and lacks dense, accurate masks for transparent regions, we extract a representative subset via uniform sampling and manually annotate high-quality masks to enable a rigorous evaluation.

We assess representative methods on this subset across three training protocols: (1) Zero-shot using official pre-trained weights, (2) Fine-tuned (FT) on our PTOD dataset, and (3) Joint FT on both PTOD and a HAMMER training subset. As shown in Table 6, while absolute performance is naturally affected by the domain shift, PolarDepth consistently maintains its advantage over RGB-only competitors. Notably, the Joint FT results demonstrate that PolarDepth possesses a significantly higher performance ceiling and superior adaptability once the target domain's physical parameters are learned.

*Table 6.* Cross-dataset generalization on the HAMMER subset. Cell values correspond to protocols: (1) Zero-shot / (2) FT on PTOD / (3) Joint FT. Results focus on transparent regions.

| Method | $\delta_1 \uparrow$ | AbsRel $\downarrow$ | RMSE $\downarrow$ | SI-Log $\downarrow$ |
|---|---|---|---|---|
| DA2 | 0.139 / 0.303 / 0.488 | 0.758 / 0.539 / 0.302 | 0.656 / 0.588 / 0.198 | 1.015 / 0.674 / 0.211 |
| Marigold | 0.040 / 0.352 / 0.497 | 0.611 / 0.528 / 0.284 | 0.398 / 0.472 / 0.182 | 0.819 / 0.710 / 0.184 |
| GW-Depth | 0.115 / 0.421 / 0.559 | 0.773 / 0.654 / 0.262 | 0.597 / 0.396 / 0.178 | 1.972 / 0.753 / 0.175 |
| MODEST | 0.022 / 0.350 / 0.433 | 0.855 / 0.688 / 0.297 | 0.796 / 0.448 / 0.218 | 4.247 / 0.851 / 0.204 |
| PolarDepth (Ours) | - / 0.420 / 0.666 | - / 0.651 / 0.179 | - / 0.347 / 0.141 | - / 0.695 / 0.168 |

