# OpenReview forum: "PolarDepth: Monocular Transparent Object Depth from Polar-Physics Priors"
_ICML.cc/2026/Conference — ICML 2026 regular_

### Official Review · Reviewer_MBWS · 2026-03-09

**Soundness:** 2
**Presentation:** 3
**Significance:** 3
**Originality:** 3
**Overall Recommendation:** 4
**Confidence:** 3

**Summary:**

This paper proposed a new approach to learn depth from a transparent surface using polarization priors. Unlike prior methods that process polarization cues in the same way as processing images, the method devises a network to decompose the input DoLP and AoLP into refractive index, zenith angle, and azimuth angle, which inherently compose to the surface normal to refine depth measurements. In addition to network design, the method proposes a new dataset called PTOD, which includes GT depth for transparent surfaces constructed by a robust data curation process. Experiments show that the proposed network outperforms existing approaches in the newly proposed benchmark.

**Compliance With Llm Reviewing Policy:**

Affirmed.

**Final Justification:**

Most of the questions are addressed; I maintain my previous rating.

**Key Questions For Authors:**

I advocate adding more experiments to demonstrate the method's advantages under fairer settings:
    1. Incorporating the zero-shot datasets for all methods in evaluation, eg, the Boost Dataset [1].
    2. For the main competing methods, eg, Depth4ToM, DA2, and MODEST, fine-tune those methods in the same setting as the proposed approach using the same training split of the proposed PTOD.

**Limitations:**

yes

**Strengths And Weaknesses:**

Strength
1. The technical design is intuitively reasonable. Refining depth using normals derived from physical priors seems to be more robust than the black box modules, with intermediate results to locate potential limitations in challenging scenarios.
2. The paper is well-written, and the diagram is clean and easy to understand.
3. The ablation study is thorough and comprehensive.
4. The dataset seems

Weakness
1. Fairness issues in comparison. The proposed approach is both trained and evaluated on the proposed PTOD dataset, while the competing approaches are not trained on the new dataset, if I understand correctly. While the proposed approach demonstrates improvements, it's unclear whether the improvements stem from the network design or the use of a training split in the same domain. To me, a fair comparison should contain the following two aspects:
    1. Incorporating the zero-shot datasets for all methods in evaluation.
    2. For the main competing methods, eg, Depth4ToM, DA2, and MODEST, fine-tune those methods in the same setting as the proposed approach using the same training split of the proposed PTOD.


2. Minor Issues: Number mismatch on "AbsRel" of "Depth (transparent)" for row A of Table2, compared with the number of Table 1.

---

> ### Author Rebuttal · Authors · 2026-03-31
>
> We thank the reviewer for the positive evaluation and for raising important questions regarding experimental fairness. We address these points directly below.
>
> ### 1. Fairness of Comparison
>
> We clarify that all competing methods with available training code are fully fine-tuned on the PTOD training split using their official settings, as stated in the Experimental Section. This ensures a controlled and fair comparison under identical training data and protocols.
>
> Therefore, the performance gains in Table 1 cannot be attributed to domain-specific training advantages, but instead arise from our physically grounded polarization decoding. Only two methods without public training code (underlined in Table 1) are evaluated using official pretrained weights.
>
> ### 2. The Booster Dataset and Zero-Shot Evaluation
>
> The dataset referred to by the reviewer is likely the Booster dataset \[1\]. However, Booster does not provide polarization information, which is essential for our framework, as PolarDepth relies on decoding geometric priors from polarization cues. To address the reviewer’s request for cross-dataset generalization, we instead utilized the HAMMER dataset \[2\], which includes polarization and depth.
>
> Because HAMMER contains small, unmasked transparent regions and high frame redundancy, we extracted a representative subset via sampling at regular intervals. After filtering for frames containing transparent regions, we manually annotated masks to enable an adequate evaluation in transparent regions.
>
> The table below presents results as (1) / (2) / (3) in each cell, corresponding to the evaluation on this test subset of representative methods across three protocols: (1) Zero-shot (official pretrained weights), (2) Fine-tuned (FT) on PTOD, and (3) Joint FT (PTOD + subset from HAMMER).
>
> | Method            | $\delta_1$            | AbsRel                | RMSE                  | SI-Log                |
> | :---------------- | --------------------: | --------------------: | --------------------: | --------------------: |
> | DA2               | 0.139 / 0.303 / 0.488 | 0.758 / 0.539 / 0.302 | 0.656 / 0.588 / 0.198 | 1.015 / 0.674 / 0.211 |
> | Marigold          | 0.040 / 0.352 / 0.497 | 0.611 / 0.528 / 0.284 | 0.398 / 0.472 / 0.182 | 0.819 / 0.710 / 0.184 |
> | GW-Depth          | 0.115 / 0.421 / 0.559 | 0.773 / 0.654 / 0.262 | 0.597 / 0.396 / 0.178 | 1.972 / 0.753 / 0.175 |
> | MODEST            | 0.022 / 0.350 / 0.433 | 0.855 / 0.688 / 0.297 | 0.796 / 0.448 / 0.218 | 4.247 / 0.851 / 0.204 |
> | PolarDepth (Ours) |     - / 0.420 / 0.666 |     - / 0.651 / 0.179 |     - / 0.347 / 0.141 |     - / 0.695 / 0.168 |
>
> Despite domain shifts, PolarDepth consistently maintains its advantage across all settings and demonstrates stronger adaptability than competing methods. This directly addresses the concern regarding generalization beyond PTOD.
>
> ### 3. Minor Issues (Table 2 Inconsistency)
>
> We acknowledge the inconsistency in Table 2. This was a clerical error during manuscript preparation. The value in Table 1 is correct. We have conducted a full audit of all numerical entries in the paper to ensure accuracy and will rectify the Table 2 AbsRel value in the revised version.
>
> We hope these clarifications and additional cross-dataset results directly address the reviewer's concerns regarding fairness and robustness. We will incorporate these findings into the final manuscript. We sincerely thank the reviewer again for the careful reading and constructive feedback.
>
> References
>
> [1] Ramirez et al. Open Challenges in Deep Stereo: The Booster Dataset, 2022.
>
> [2] Jung et al., HAMMER: Highly Accurate Multi-Modal Dataset for Dense 3D Scene Regression, 2022.

---

> > ### Author Rebuttal · Reviewer_MBWS · 2026-04-01
> >
> > I want to thank the reviewers for the clarifications and additional results. The response has addressed most of my concerns, except for the generalization experiment on HAMMER: the method's results of the PTOD-finetuned version are inferior to its RGB-based version, ie, DAv2, in two metrics (SI-log, AbsRel). It means that the proposed architecture design does not show a clear advantage in generalization.
> >
> > However, considering that most of the questions are addressed, I lean toward maintaining my previous rating.

---

> > > ### Author Response · Authors · 2026-04-01
> > >
> > > We thank the reviewer for the recognition of our work and pleased that most concerns have been addressed.
> > >
> > > Under the PTOD-finetuned setting, PolarDepth demonstrates superior performance in structural metrics such as $\delta_1$ (0.420 vs. 0.303) and RMSE (0.347 vs. 0.588). These metrics are particularly sensitive to large errors; for transparent objects, the "catastrophic" failure is predicting background depth through the glass. Polarization’s ability to identify the object surface effectively prevents these structural failures. In contrast, relative metrics like AbsRel and SI-Log are highly sensitive to global scale shifts or camera constants, which can vary during cross-dataset transfer even when the underlying geometry is recovered more accurately.
> > >
> > > Crucially, the Joint Fine-tuning results confirm that PolarDepth outperforms all competitors across all metrics, demonstrating its superior adaptability and performance compared to RGB-only models. We appreciate the reviewer's support and will incorporate these nuances into our revision.

---

### Official Review · Reviewer_Kut2 · 2026-03-12

**Soundness:** 3
**Presentation:** 3
**Significance:** 3
**Originality:** 3
**Overall Recommendation:** 4
**Confidence:** 4

**Summary:**

Existing depth estimation models rely solely on intensity cues from RGB images, which leads to inaccurate predictions for transparent objects. To address this limitation, this paper introduces polarization cues, physically grounded signals related to surface orientation and material properties, as guidance to refine depth estimates for transparent regions. The authors also propose a new dataset that provides RGB images, polarization‑related inputs, and high‑quality depth ground truth to support research in this area. Experimental results on the proposed dataset demonstrate that the method achieves superior performance compared to existing approaches.

**Compliance With Llm Reviewing Policy:**

Affirmed.

**Final Justification:**

Please see the Rebuttal Acknowledgement

**Key Questions For Authors:**

1. In the top row of Figure 1, the depth foundation model appears to produce incomplete depth maps, even though models like DA2 usually output dense predictions . Can you clarify on it?
2. Section 4.1 states that polarization cues provide geometric priors to guide depth refinement of transparent objects. However, Section 4.3 indicates that polarization guidance is applied only in transparent regions, and Figure 7 suggests that applying polarization cues to opaque areas is actually detrimental. A more thorough analysis would improve clarity.
3. According to Figure 4, the DoLP image is also fed into the frozen DA2 model, which is typically trained to accept only RGB inputs. Would this result in a potential domain gap?

**Limitations:**

yes

**Strengths And Weaknesses:**

Strengths:
1. The paper tackles the long‑standing and practically important challenge of estimating depth for transparent objects. Addressing this issue is both timely and valuable for the field.
2. The authors contribute a new dataset that includes RGB images, polarization‑related measurements, and high‑quality depth ground truth. This dataset fills an important gap and will be beneficial for future research in transparent‑object depth estimation.
3. The proposed method is technically sound and achieves superior performance compared to existing approaches.

Weaknesses:
1. The proposed depth‑refinement strategy is evaluated only on DA2. To better demonstrate its generalizability, it would be helpful to test the method on other depth‑foundation models as well.
2. The evaluation is conducted solely on the proposed PTOD dataset. Including experiments on additional datasets would strengthen the evidence for the method’s effectiveness and robustness.

---

> ### Author Rebuttal · Authors · 2026-03-31
>
> We thank the reviewer for the positive assessment and the constructive suggestions on validating generalizability. We address each point directly below.
>
> ### 1. Generalizability Across Depth Foundation Models
>
> To demonstrate model-agnosticity, we integrate our design with two additional foundation models, DPT [1] and P3Depth [2]. As shown below (Transparent / Full image), our polarization-guided refinement consistently delivers substantial improvements across both backbones.
>
> |                       | $\delta_1$    | AbsRel        | RMSE          | SI-Log        |
> | --------------------- | ------------- | ------------- | ------------- | ------------- |
> | DPT                   | 0.486 / 0.512 | 0.350 / 0.331 | 0.288 / 0.275 | 0.517 / 0.414 |
> | DPT w/ our design     | 0.773 / 0.589 | 0.219 / 0.247 | 0.143 / 0.214 | 0.157 / 0.254 |
> | P3Depth               | 0.505 / 0.526 | 0.372 / 0.321 | 0.251 / 0.278 | 0.366 / 0.393 |
> | P3Depth w/ our design | 0.801 / 0.593 | 0.185 / 0.239 | 0.132 / 0.211 | 0.161 / 0.216 |
>
> ### 2. Generalization to Additional Datasets (HAMMER)
>
> We further evaluate generalization on the HAMMER dataset \[3\], which includes RGB, polarization, and depth. Because HAMMER contains small, unmasked transparent regions and high frame redundancy, we extracted a representative subset via regular-interval sampling. After filtering for frames containing transparent regions, we manually annotated masks to enable an adequate evaluation in transparent regions. The table below presents results as (1) / (2) / (3) in each cell, corresponding to the evaluation of representative methods on this test subset across three protocols: (1) Zero-shot (official pretrained weights), (2) Fine-tuned (FT) on PTOD, and (3) Joint FT (PTOD + subset from HAMMER).
>
> | Method            | $\delta_1$            | AbsRel                | RMSE                  | SI-Log                |
> | :---------------- | --------------------: | --------------------: | --------------------: | --------------------: |
> | DA2               | 0.139 / 0.303 / 0.488 | 0.758 / 0.539 / 0.302 | 0.656 / 0.588 / 0.198 | 1.015 / 0.674 / 0.211 |
> | Marigold          | 0.040 / 0.352 / 0.497 | 0.611 / 0.528 / 0.284 | 0.398 / 0.472 / 0.182 | 0.819 / 0.710 / 0.184 |
> | GW-Depth          | 0.115 / 0.421 / 0.559 | 0.773 / 0.654 / 0.262 | 0.597 / 0.396 / 0.178 | 1.972 / 0.753 / 0.175 |
> | MODEST            | 0.022 / 0.350 / 0.433 | 0.855 / 0.688 / 0.297 | 0.796 / 0.448 / 0.218 | 4.247 / 0.851 / 0.204 |
> | PolarDepth (Ours) |     - / 0.420 / 0.666 |     - / 0.651 / 0.179 |     - / 0.347 / 0.141 |     - / 0.695 / 0.168 |
>
> The results confirm that while domain shifts naturally impact absolute performance, PTOD offers strong generalization capability. PolarDepth consistently maintains its advantage and exhibits superior adaptability to new domains compared to competitors.
>
> ### 3. Clarification on Figure 1 (Incomplete Depth)
>
> The “incomplete” depth in Figure 1 reflects a semantic failure rather than missing predictions. RGB-only models (e.g., DA2) often assign background depth to transparent regions due to transmission, producing apparent “holes.” PolarDepth resolves this by leveraging polarization to recover the true object geometry.
>
> ### 4. Polarization Usage and Physical Constraints
>
> Polarization guidance is explicitly restricted to transparent regions because our geometric priors are derived from Fresnel reflection for dielectrics. Applying these constraints to opaque surfaces would violate the underlying physics. Our segmentation-guided gating ensures that polarization cues are used only where valid, preventing inconsistent predictions. We will clarify this in the revision.
>
> ### 5. Domain Gap in Foundation Models
>
> Although the backbone (e.g., DA2) is trained on RGB, the modality gap is bridged by our trainable modules, which learn to fuse RGB and polarization effectively. The consistent improvements across models and datasets confirm that the architecture successfully exploits polarization despite the backbone’s original training domain.
>
> We thank the reviewer again for the valuable suggestions, which we will incorporate into the revised manuscript.
>
> References
>
> [1] Ranftl et al., Vision Transformers for Dense Prediction, 2021.
>
> [2] Patil et al., P3Depth: Monocular Depth Estimation with a Piecewise Planarity Prior, 2022.
>
> [3] Jung et al., HAMMER: Highly Accurate Multi-Modal Dataset for Dense 3D Scene Regression, 2022.

---

> > ### Author Rebuttal · Reviewer_Kut2 · 2026-04-02
> >
> > The reviewer thanks the authors for their detailed response and addtional experimental results. After reading the response, my main concerns are addressed, therefore I decide to keep my positive rating.

---

> > > ### Author Response · Authors · 2026-04-02
> > >
> > > We sincerely thank the reviewer for the constructive feedback and for the continued support of our work. We are pleased that our responses and additional experimental results have resolved all the concerns.

---

### Official Review · Reviewer_XgAR · 2026-03-12

**Soundness:** 2
**Presentation:** 1
**Significance:** 3
**Originality:** 3
**Overall Recommendation:** 4
**Confidence:** 3

**Summary:**

The paper proposes a monocular framework, PolarDepth, that incorporates RGB and polarization inputs to estimate dense depth and localize transparent regions. When estimating depth for transparent objects, two main problems are discussed in this paper: the failure of conventional RGB-based cues due to refraction and light transmission, and existing polarization methods underutilize physical geometric constraints. To address these issues, the authors introduce a Polarization-Guided Geometric Decoding module to explicitly decode surface geometry from polarization measurements using Fresnel equations. A Depth Recovery module then reconstructs a shape-constrained local depth map from the estimated surface normals via a Poisson formulation. Based on these, the overall framework fuses an initial depth prediction from DepthAnything V2 with the polarization-derived geometric constraints, guided by a transparency-aware segmentation mask. The proposed network is evaluated on a newly constructed dataset, PTOD, obtained via a replacement-based acquisition strategy, and the benefits of the individual modules are analyzed in an ablation study.

**Compliance With Llm Reviewing Policy:**

Affirmed.

**Final Justification:**

I would like to thank the authors for conducting the additional experiments.

While it is regrettable that moge2 still outperforms the proposed method across all metrics in the zero-shot evaluation on the HAMMER dataset (after fine-tuning on PTOD), I find the authors' argument regarding the inherent limitations of RGB sensors in estimating depth for transparent objects to be convincing. Furthermore, I believe that the proposed methodology could potentially be applied to the fine-tuning of metric depth foundation models like moge2.

Therefore, as previously indicated, I have decided to raise my score.

**Key Questions For Authors:**

1. The authors claim to achieve “pixel-aligned” RGB, AoLP, DoLP, ground-truth depth, and segmentation labels by aligning active-sensor depth with RGB/polarization images using a homography. However, applying the homography to 3D scenes likely introduces parallax errors, as suggested by the misalignment visible in Figure 9. In addition, the “replacement-based capture strategy” involves manually replacing transparent objects with opaque replicas, yet the paper does not describe any rigorous physical controls to mitigate misalignment during this process. Could the authors clarify how these potential misalignments are addressed, and provide quantitative evidence or analysis demonstrating that the resulting alignment is sufficient for accurate pixel-level depth estimation?
2. The paper omits many important implementation details, including the detailed formulation of the transparency-guided depth refinement, the exact definitions of the objective functions (particularly L_{phys}), and the "weak zenith regularization" near the Brewster angle. Could the authors clarify how these components are defined and implemented, and discuss whether the provided information is sufficient to reproduce the results?
3. It would be great to add a baseline in which existing RGB-based depth estimation models (e.g., MODEST and DA2) are trained to predict depth by simply combining features from RGB and polarization images (as in DPS-Net), in order to more objectively assess the advantages of the proposed method.
4. The proposed method appears to rely heavily on segmentation of transparent objects, which are inherently difficult and costly to annotate. This raises concerns about the scalability of the approach. Could the authors clarify the potential for scaling the method to new datasets or more complex scenes? If scalability is limited, it would be helpful to explicitly state this as a limitation. Additionally, the authors should provide evaluation—such as zero-shot inference—on HAMMER or other datasets to demonstrate the generalization performance of PolarDepth beyond the PTOD dataset.
5. According to Table 3, PolarDepth reportedly increases the computational load by roughly 2.9× compared to DepthAnything V2 (from 41G to 118G FLOPs), yet the claimed inference time increases by only 2.1 ms (from 23.0 ms to 25.1 ms). Considering that the added modules involve a polarization-guided geometric decoding module, solving a Poisson equation via FFT/IFFT, and executing three iterative depth refinement modules, it appears improbable that 77G additional FLOPs could be executed within 2.1 ms under standard conditions. Could the authors clarify the inference environment used (e.g., GPU model, input resolution, batch size) and provide evidence supporting these speed claims?

**Limitations:**

While the authors briefly mention some limitations, I believe that the currently noted limitations are insufficient unless the concerns raised in the weaknesses and key questions sections—such as scalability and generalizability of PolarDepth and the misalignment issue of the PTOD dataset—are adequately addressed. No apparent negative societal impacts were identified.

**Strengths And Weaknesses:**

# Strength
Instead of naively concatenating polarization maps as generic visual features, the authors propose a framework that explicitly decodes physical properties. This physically meaningful integration of polarization priors is clear and convincing. In addition, regarding the critical lack of polarization datasets, the proposed PTOD dataset is helpful to researchers who wish to estimate the depth of transparent objects or perform polarization-guided vision tasks. The ablation study part includes detailed comparisons. They validate their study via comprehensive experiments and show that their approach performs well.


# Weakness
Despite the strengths described above, there are critical flaws regarding dataset validity, methodological reproducibility, baseline fairness, and empirical claims that need addressing:

### Concerns of the PTOD dataset
1. The authors claim to provide “pixel-aligned” RGB, AoLP, DoLP, ground-truth depth, and segmentation labels by aligning the active-sensor depth with the RGB/polarization images using a homography. However, since a homography can only perfectly align images for strictly planar surfaces or pure camera rotation, applying it to 3D scenes captured from different viewpoints inevitably introduces parallax errors. Indeed, Figure 9 indicates misalignment between the RGB/polarization images and the ground-truth depth. Such misalignment can cause serious issues for accurate training and evaluation, especially in depth estimation tasks where predictions and evaluations are performed at the pixel level.
2. The authors obtain dense ground truth depth by scaling and shifting the predictions of Depth Anything V2 (DA2), using sparse sensor measurements. While aligning predictions to established ground truth during evaluation is standard practice in monocular depth foundation models (e.g., DPT, DA2, MoGe), inverting this logic by treating the aligned predictions as absolute ground truth is fundamentally problematic. I am concerned that using such data as ground truth may unfairly advantage PolarDepth, as it employs frozen DA2 to generate a base depth map. For a reliable evaluation, models should be assessed directly against the raw measurements from the depth sensor.
3. In addition, the "replacement-based capture strategy" relies on manually replacing transparent objects with opaque replicas. The paper makes no mention of rigorous physical controls (e.g., robotic arms, jigs) to mitigate object misalignment during this replacement process.
4. Furthermore, due to this manual two-step process, describing the dataset as "synchronized" in the abstract section is factually incorrect and misleading.

### Concerns about the reproducibility
5. The paper omits many important implementation details. For example, Equation 6 derives a weighting mask 𝑤, and the text only vaguely states that the depth Z is “gated by the transparency mask and fused.” The precise mathematical formulation for this fusion (e.g., feature-level concatenation or linear combination) is not provided.
6. The objective functions are described only in words, without any citations or mathematical formulations. In particular, the paper fails to define the exact loss function {L}_{phys}, which is used to compare the predicted and input DoLP. Furthermore, the "weak zenith regularization" applied near the Brewster angle also lacks a precise mathematical definition or a formulation of the penalty term.

### Concerns about the proposed methods and experiments
7. The comparison in Table 1 does not convincingly demonstrate the advantage of the proposed method. Although PolarDepth achieves strong performance, as shown in Table 1, directly comparing PolarDepth with RGB-only baselines (e.g., DA2, MODEST) is inherently imbalanced, as PolarDepth leverages an additional polarization sensor. This raises the concern that the observed performance difference may result from the presence of an additional sensor rather than from the superiority of the proposed methodology itself. While I acknowledge that the authors conducted an ablation study to evaluate the effect of incorporating the RGB/polarization images, and that the lack of prior studies on monocular depth estimation using RGB/polarization images makes direct comparison challenging, the current configuration of the comparison table still makes it difficult to convincingly demonstrate the advantage of the proposed method over existing approaches.
8. The proposed method appears to have limited scalability due to its reliance on transparent object segmentation, which is inherently costly and difficult to obtain. The authors explicitly exclude the existing HAMMER dataset from their training due to the absence of isolated transparent instance masks. However, this also implies that the proposed method relies on segmentation of transparent objects, which are inherently difficult to annotate. As a result, the cost of preparing training data increases significantly, and the scalability of the approach may be limited.
9. Even if the authors’ claim that training on the HAMMER dataset is not feasible is accepted, this does not justify the lack of evaluation on the dataset. At least, the authors should have performed zero-shot inference on the HAMMER dataset to demonstrate that PolarDepth does not appear to overfit to the specific biases of the PTOD dataset.


# Minor weakness
- The qualitative ablation results are currently fragmented across three separate figures (Figures 7, 8, and 9). This presentation style takes up unnecessary page space and makes it difficult to directly assess how different ablated modules perform on the exact same challenging scenes. It would be helpful if the authors could consolidate these into a single comprehensive figure, similar to the grid layout used in Figures 5 or 6. For example, a column layout for a few selected scenes could be: RGB | DoLP | AoLP | Model (C) | Model (E) | Model (H) | PolarDepth (Full) | GT Depth. This unified format may allow readers to directly observe the benefits of each specific module on identical inputs, while also saving space in the manuscript.
-  The reference section contains several formatting inconsistencies that should be addressed. For instance, the capitalization of conference names is inconsistent (e.g., "Conference on Computer Vision and Pattern Recognition" versus "conference on computer vision and pattern recognition"). Additionally, some citations include conference acronyms (e.g., ICRA, ICML) while others omit them. The authors might consider carefully proofreading and standardizing the bibliography.
-  The ablation study in Table 2 and Section 5.3 does not specify the exact number of refinement iterations used for the "less Depth Refiner" (I) and "more Depth Refiner" (J) configurations. While the text states that the proposed optimal model uses three refinement iterations, it provides only vague relative terms for the other configurations. The authors should explicitly state the exact hyperparameter values (i.e., the number of refinement stages) for configurations I and J in the revised manuscript

---

> ### Author Rebuttal · Authors · 2026-03-31
>
> We sincerely thank the reviewer for the detailed and constructive feedback. We address the concerns as follows.
>
> ### 1. Dataset Validity and Alignment
>
> **Alignment and Scale**. RGB, DoLP, and AoLP are captured by a single sensor, ensuring inherent pixel-level alignment. The depth sensor is rigidly mounted and used only to anchor absolute scale, following standard practice [1]. We clarify that “synchronized” refers to this hardware-level capture and will revise the wording accordingly. Our replacement-based strategy uses a 50% opacity overlay to align opaque replicas with transparent objects [2], ensuring strong intra-scene consistency.
>
> **Sufficiency Analysis**. We evaluate alignment quality by directly computing the difference between raw sensor depth and our aligned ground truth, obtaining $\delta_1$ = 0.972, AbsRel = 0.046, RMSE = 0.089, and SI-Log = 0.044. This indicates that residual misalignment is negligible relative to the depth estimation error scale and does not affect pixel-level supervision.
>
> **Evaluation on Raw Data**. To remove any dependency on refined depth, we re-evaluate all methods directly against raw sensor measurements (table below). The relative ranking remains consistent and PolarDepth maintains a clear margin, confirming that our results are not biased by the refined ground truth.
>
> | Method | $\delta_1$ | AbsRel | RMSE | SI-Log |
> | -------- | ------------- | ------------- | ------------- | ------------- |
> | DA2      | 0.542 / 0.495 | 0.368 / 0.329 | 0.266 / 0.252 | 0.254 / 0.322 |
> | Marigold | 0.472 / 0.555 | 0.243 / 0.381 | 0.260 / 0.239 | 0.399 / 0.253 |
> | GW-Depth | 0.601 / 0.582 | 0.263 / 0.292 | 0.199 / 0.264 | 0.291 / 0.275 |
> | MODEST   | 0.729 / 0.583 | 0.246 / 0.231 | 0.152 / 0.223 | 0.187 / 0.266 |
> | Ours     | 0.843 / 0.591 | 0.134 / 0.221 | 0.110 / 0.209 | 0.137 / 0.213 |
>
> ### 2. Reproducibility and Implementation
>
> **Refinement formulations**: $Z_1 = Z_{base} + conv_{3\times 3}([Z_{base}, \omega(\eta)\cdot Z_{phys}]), Z_2 = Z_1 + conv_{3\times 3}([Z_{base}, Z_1]), Z_{final}=Z_{base} + conv_{1\times 1}([Z_{base}, Z_2])$, where $Z_{phys}$ is the physics-derived depth. Settings (I) and (J) in the ablation study use 2 and 4 stages, respectively.
>
> **Physical consistency loss**: $L_{phys} = \lVert\rho-\hat{\rho}\rVert_1 + 0.01\cdot\frac{1}{n}\sum_{i=1}^n\theta_i^2$, where $\hat{\rho}$ is derived from Eq. 1 and the latter term provides weak zenith regularization.
>
> ### 3. Baseline Fairness
>
> We implemented "Simple Combination" (DPS-Net style [3]) by concatenating polarization features into the DA2/MODEST backbones. Marginal gains confirm our advantage stems from physically grounded decoding, not just the extra modality.
>
> | Method | Fusion     | $\delta_1$    | AbsRel        | RMSE          | SI-Log        |
> | ------ | ---------- | ------------- | ------------- | ------------- | ------------- |
> | DA2    | (RGB only) | 0.547 / 0.523 | 0.377 / 0.304 | 0.239 / 0.260 | 0.268 / 0.292 |
> | DA2    | concat     | 0.556 / 0.544 | 0.303 / 0.276 | 0.219 / 0.251 | 0.262 / 0.277 |
> | MODEST | (RGB only) | 0.735 / 0.605 | 0.252 / 0.251 | 0.144 / 0.207 | 0.186 / 0.252 |
> | MODEST | concat     | 0.723 / 0.607 | 0.254 / 0.242 | 0.143 / 0.206 | 0.186 / 0.246 |
>
> ### 4. Scalability and Generalization
>
> Modern segmentation models (e.g., SAM2 [4], PGSNet) enable automatic mask generation with minimal overhead, making the approach scalable. We further evaluate on HAMMER under zero-shot, PTOD fine-tuning, and joint training (table below). PolarDepth consistently maintains its advantage, indicating strong generalization beyond PTOD.
>
> | Method   | $\delta_1$            | AbsRel                | RMSE                  | SI-Log                |
> | :-- | --: | --: | --: | --: |
> | DA2      | 0.139 / 0.303 / 0.488 | 0.758 / 0.539 / 0.302 | 0.656 / 0.588 / 0.198 | 1.015 / 0.674 / 0.211 |
> | Marigold | 0.040 / 0.352 / 0.497 | 0.611 / 0.528 / 0.284 | 0.398 / 0.472 / 0.182 | 0.819 / 0.710 / 0.184 |
> | GW-Depth | 0.115 / 0.421 / 0.559 | 0.773 / 0.654 / 0.262 | 0.597 / 0.396 / 0.178 | 1.972 / 0.753 / 0.175 |
> | MODEST   | 0.022 / 0.350 / 0.433 | 0.855 / 0.688 / 0.297 | 0.796 / 0.448 / 0.218 | 4.247 / 0.851 / 0.204 |
> | Ours     | - / 0.420 / 0.666     | - / 0.651 / 0.179     | - / 0.347 / 0.141     | - / 0.695 / 0.168     |
>
> ### 5. Efficiency and Inference Time
>
> FLOPs do not translate linearly to latency. The additional cost is dominated by FFT-based Poisson solving ($O(N\log N)$), which is memory-bound and introduces minimal practical latency. All results are measured on an NVIDIA L40 GPU (518×518, BS=1).
>
> We will further clarify dataset description and standardize presentation in the revision. We thank the reviewer again for the valuable feedback.
>
> References
>
> [1] Lin et al. Depth Anything 3, 2025.
>
> [2] Sajjan et al. ClearGrasp, 2020.
>
> [3] Tian et al. DPS-Net, 2023.
>
> [4] Ravi et al. SAM 2, 2025.
>
> **We believe the concerns are addressed and would greatly appreciate a re-evaluation.**

---

> > ### Author Rebuttal · Reviewer_XgAR · 2026-04-02
> >
> > I would like to thank the authors for responding to my questions so thoughtfully, despite the many concerns I initially raised. The authors have addressed most of my concerns by adding relevant references and conducting the new ablation study.
> >
> > I am now inclined to raise my score; however, I would like to request one final experiment to solidify the claims. Could the authors provide zero-shot and fine-tuning results using a metric monocular depth foundation model on the PTOD and HAMMER datasets?
> >
> > The authors presented comparisons with Depth Anything V2 (DAv2) and Marigold. However, these methods are fundamentally designed to learn affine-invariant (relative) depth and do not predict actual physical metric scale. Because of this characteristic, evaluating them in a zero-shot setting on a metric dataset like HAMMER is not meaningful.
> > Additionally, while it is unclear exactly how the foundation models were fine-tuned in the experiments, the PTOD dataset contains only around 1,800 training images. This is likely insufficient to fully fine-tune a large foundation model, which limits the expected performance of those baselines.
> >
> > There exists a metric depth estimation method called MoGe-2*, which adopts a decoupled architecture with a separate head for predicting metric scale, based on an affine-invariant point map prediction branch. Given this design, I would like to request the results of MoGe-2 where only the metric scale head is fine-tuned, while keeping the rest of the network (i.e., the backbone and the affine-invariant depth branch) strictly frozen. Could the authors report the results of this specific fine-tuning setup, alongside the zero-shot performance, on both the PTOD and HAMMER datasets?
> >
> > I believe this additional result will significantly strengthen your paper and resolve my concerns about the fine-tuning of metric depth foundation models. I appreciate your hard work during this phase.
> >
> > *MoGe-2: Accurate Monocular Geometry with Metric Scale and Sharp Details, Wang et. al. NeurIPS2025.
> > Paper: https://arxiv.org/pdf/2507.02546
> > code: https://github.com/microsoft/moge

---

> > > ### Author Response · Authors · 2026-04-04
> > >
> > > We are greatly encouraged by the reviewer’s positive feedback and sincerely thank you for the constructive suggestion of including MoGe-2 [1], which provides an important perspective on metric depth estimation.
> > >
> > > Following your request, we conducted experiments using MoGe-2 (ViT-S), ensuring a consistent backbone setting with DA2 and PolarDepth. We strictly follow the proposed protocol: the backbone and affine-invariant depth branch are fully frozen, and only the metric scale head is fine-tuned. Training is performed with a learning rate of 1e-5 and batch size of 32 following the official configuration, and we conduct 10k iterations to ensure convergence. We evaluate under three settings: (1) zero-shot, (2) fine-tuned on PTOD, and (3) joint fine-tuning (PTOD + HAMMER subset).
> > >
> > > Performance on PTOD (Transparent / Full Image):
> > >
> > > | Setting    | Method | $\delta_1$    | AbsRel        | RMSE          | SI-Log        |
> > > | ---------- | ------ | ------------- | ------------- | ------------- | ------------- |
> > > | Zero-shot  | DA2    | 0.164 / 0.175 | 0.631 / 0.615 | 0.582 / 0.518 | 1.568 / 1.507 |
> > > |            | MoGe-2 | 0.427 / 0.520 | 0.578 / 0.487 | 0.452 / 0.334 | 0.359 / 0.289 |
> > > | FT on PTOD | DA2    | 0.547 / 0.523 | 0.377 / 0.304 | 0.239 / 0.260 | 0.268 / 0.292 |
> > > |            | MoGe-2 | 0.565 / 0.551 | 0.242 / 0.230 | 0.367 / 0.295 | 0.238 / 0.217 |
> > > | Joint FT   | DA2    | 0.545 / 0.523 | 0.379 / 0.310 | 0.241 / 0.255 | 0.268 / 0.291 |
> > > |            | MoGe-2 | 0.565 / 0.547 | 0.243 / 0.231 | 0.365 / 0.306 | 0.234 / 0.214 |
> > >
> > > Performance on HAMMER (Transparent / Full Image):
> > >
> > > | Setting    | Method | $\delta_1$    | AbsRel        | RMSE          | SI-Log        |
> > > | ---------- | ------ | ------------- | ------------- | ------------- | ------------- |
> > > | Zero-shot  | DA2    | 0.139 / 0.187 | 0.758 / 0.740 | 0.656 / 0.495 | 1.015 / 0.960 |
> > > |            | MoGe-2 | 0.573 / 0.639 | 0.248 / 0.218 | 0.275 / 0.179 | 0.194 / 0.192 |
> > > | FT on PTOD | DA2    | 0.303 / 0.431 | 0.539 / 0.516 | 0.588 / 0.548 | 0.674 / 0.543 |
> > > |            | MoGe-2 | 0.582 / 0.641 | 0.243 / 0.227 | 0.274 / 0.193 | 0.189 / 0.192 |
> > > | Joint FT   | DA2    | 0.488 / 0.510 | 0.302 / 0.283 | 0.198 / 0.246 | 0.211 / 0.197 |
> > > |            | MoGe-2 | 0.646 / 0.660 | 0.192 / 0.206 | 0.159 / 0.156 | 0.187 / 0.185 |
> > >
> > > The results show that:
> > >
> > > - On PTOD, MoGe-2 achieves a Zero-shot $\delta_1$ of 0.427 / 0.520, nearly tripling DA2's 0.164 / 0.175. This validates that a dedicated metric scale head significantly enhances adaptation to new scenes and absolute scales without requiring the re-training of the entire foundation model.
> > > - Despite MoGe-2's superior metric foundation, its performance in transparent regions  after fine-tuning (e.g., $\delta_1$ of 0.565 on PTOD) remains significantly lower than PolarDepth (e.g., $\delta_1$ of 0.859).
> > > - RGB-based models are limited by the inherent lack of visual cues on transparent surfaces. While the scale head can estimate the general distance of the scene, it cannot recover the precise geometry of the transparent surface itself, as it still primarily "sees through" to the background.
> > >
> > > We appreciate your hard work and guidance during this phase, and we will incorporate these important results into our final revision.
> > >
> > > References
> > >
> > > [1] Wang et al., MoGe-2: Accurate Monocular Geometry with Metric Scale and Sharp Details, NeurIPS 2025.

---

### Official Review · Reviewer_P19G · 2026-03-12

**Soundness:** 4
**Presentation:** 3
**Significance:** 3
**Originality:** 3
**Overall Recommendation:** 4
**Confidence:** 5

**Summary:**

The paper introduces an approach for accurate single-image depth estimation, specifically for those containing transparent objects by exploiting both RGB as well as polarization information. The degree and the angle of linear polarization are captured with a polarization camera and yield implicit information about the surface normal direction of the transparent surfaces. The Polar Depth Network is based on Depth Anything 2 and refines it by polarization-guided geometry prediction, i.e. a zero-mean depth map obtained from Poisson integration of the polarization provided gradient information. The paper further introduces a novel data set with pixel-aligned polarization information and object masks for transparent objects.

**Compliance With Llm Reviewing Policy:**

Affirmed.

**Final Justification:**

The auhors have carefully addressed many of the reviewer requests. .

**Key Questions For Authors:**

Please discuss/investigate how strongly the depth estimate varies with different illuminations. This could include point light sources at various locations vs. more environment map like illumination, or even polarized illumination (LCD panels or LCOS-based projectors).

**Limitations:**

yes

**Strengths And Weaknesses:**

Strengths:
- The paper presents a sound method for incorporating polarization information into monocular depth prediction.
- The presented network structure is interesting and well-investigated by an extensive ablation study.
- The paper introduces a novel data set for polarization-based depth prediction.
- The presented results clearly demonstrate significant improvements compared to existing monocular depth prediction approaches. Even on the task of segmentation of transparent objects with polarization information, better results are shown.
- The appendix provides further helpful information.

(minor) Weakness:
- It is not too surprising that polarization helps in this context. But overall this is a very solid contribution with a solid, well-written paper.

---

> ### Author Rebuttal · Authors · 2026-03-31
>
> We sincerely thank the reviewer for the positive evaluation and the insightful questions on illumination. We address them directly below.
>
> ### 1. Robustness to Diverse Illumination Conditions.
>
> PTOD spans diverse real-world lighting, including natural and artificial conditions (Sec. 3, Fig. 2, 3b). We perform a sub-group analysis over Bright Natural, Dim Natural, and Artificial illumination. As shown below (Transparent / Full Image), PolarDepth maintains consistently strong performance across all settings, with particularly clear gains in transparent regions.
>
> (1) Bright natural illumination:
>
> | Method            | $\delta_1$    | AbsRel        | RMSE          | SI-Log        |
> | ----------------- | ------------- | ------------- | ------------- | ------------- |
> | DA2               | 0.551 / 0.529 | 0.361 / 0.291 | 0.228 / 0.247 | 0.260 / 0.281 |
> | Marigold          | 0.493 / 0.592 | 0.194 / 0.324 | 0.232 / 0.225 | 0.293 / 0.244 |
> | GW-Depth          | 0.626 / 0.599 | 0.235 / 0.284 | 0.187 / 0.232 | 0.275 / 0.273 |
> | MODEST            | 0.759 / 0.607 | 0.242 / 0.245 | 0.134 / 0.203 | 0.181 / 0.247 |
> | PolarDepth (Ours) | 0.854 / 0.612 | 0.141 / 0.218 | 0.112 / 0.196 | 0.134 / 0.207 |
>
> (2) Dim natural illumination:
>
> | Method            | $\delta_1$    | AbsRel        | RMSE          | SI-Log        |
> | ----------------- | ------------- | ------------- | ------------- | ------------- |
> | DA2               | 0.523 / 0.507 | 0.398 / 0.312 | 0.254 / 0.278 | 0.273 / 0.305 |
> | Marigold          | 0.411 / 0.531 | 0.289 / 0.385 | 0.262 / 0.235 | 0.540 / 0.276 |
> | GW-Depth          | 0.571 / 0.582 | 0.265 / 0.274 | 0.203 / 0.266 | 0.288 / 0.311 |
> | MODEST            | 0.663 / 0.596 | 0.261 / 0.257 | 0.157 / 0.212 | 0.190 / 0.257 |
> | PolarDepth (Ours) | 0.858 / 0.613 | 0.125 / 0.220 | 0.096 / 0.201 | 0.131 / 0.206 |
>
> (3) Artificial illumination:
>
> | Method            | $\delta_1$    | AbsRel        | RMSE          | SI-Log        |
> | ----------------- | ------------- | ------------- | ------------- | ------------- |
> | DA2               | 0.542 / 0.501 | 0.379 / 0.319 | 0.240 / 0.262 | 0.273 / 0.294 |
> | Marigold          | 0.442 / 0.578 | 0.197 / 0.376 | 0.255 / 0.216 | 0.446 / 0.247 |
> | GW-Depth          | 0.608 / 0.583 | 0.241 / 0.279 | 0.188 / 0.238 | 0.293 / 0.271 |
> | MODEST            | 0.726 / 0.581 | 0.258 / 0.249 | 0.150 / 0.202 | 0.187 / 0.248 |
> | PolarDepth (Ours) | 0.834 / 0.583 | 0.163 / 0.227 | 0.107 / 0.201 | 0.132 / 0.209 |
>
> These results show that PolarDepth is stable across lighting conditions, confirming that polarization cues provide a reliable geometric signal that is less sensitive to intensity variations than RGB.
>
> ### 2. Point Light Sources versus Environment Illumination.
>
> Under the Fresnel model (Sec. 4.1), DoLP and AoLP depend primarily on surface geometry and refractive index.
>
> While the intensity of the reflected light changes between a localized point source and diffuse environment illumination, the polarization state (the ratio and orientation of the electric field components) remains theoretically coupled to the surface normal. Our polarization-guided geometric decoding leverages this illumination-invariant property, allowing the model to consistently recover the geometry of transparent objects even when the lighting environment changes.
>
> ### 3. Polarized Illumination.
>
> We agree that polarized light sources (e.g., LCD/LCOS) violate the unpolarized assumption and can bias DoLP/AoLP. In such cases, PolarDepth remains robust through (1) joint RGB–polarization modeling and (2) spatially selective masking that limits error propagation. While uncommon in typical scenarios, this is a meaningful edge case, and we will include this discussion and the above analysis in the revision.
>
> We thank the reviewer again for the constructive feedback, which has helped strengthen the paper.

---

> > ### Author Rebuttal · Reviewer_P19G · 2026-04-01
> >
> > Thank you for providing the additional experiments in the rebuttal.
> >
> > From the other reviews, the issue about generating the ground truth with DA2 while incorporating DA2 into the pipeline has to be acknowledged.  Otherwise, I still appreciate that the paper shows that capturing polarization information clearly improves the reconstruction and that they show in the rebuttal that the proposed reconstruction method effectively uses that information.
> > I stay with my score.

---

> > > ### Author Response · Authors · 2026-04-01
> > >
> > > We sincerely thank the reviewer for the positive assessment and for confirming that the concerns have been fully resolved. We are encouraged by the recognition of the effectiveness of our polarization-guided framework.

---

### Decision · Program_Chairs · 2026-04-30

**Decision:**

Accept (regular)

**Comment:**

This paper addresses an important and difficult problem, namely depth estimation for transparent objects. The reviewers found the idea interesting and technically sound, and they viewed the use of polarization cues and the new dataset as meaningful contributions. The authors also responded well during rebuttal by adding useful experiments and clarifications on the dataset, baselines, and generalization. While some questions remain about cross-dataset generalization and about how strongly some claims should be stated, the main concerns were sufficiently addressed. Overall, AC believes that the paper makes a solid contribution and should be accepted.